# LLM Priors for ERM over Programs

**Shivam Singhal** [* 1]  **Priyadarsi Mishra** [* 1]  **Eran Malach** [2]  **Tomer Galanti** [1]

## Abstract

We study program-learning methods that are efficient in both samples and computation. Classical learning theory suggests that when the target admits a short program description, for example a short piece of "Python code", it can be learned from few examples by ERM over the program class. However, this approach relies on enumerating candidate programs, which is typically exponential in the description length; gradient-based training avoids this explicit search but, for some families of short programs, can require exponentially many samples to succeed. We propose LLM-PV, a propose-and-verify recipe that enables ERM-style selection over a discrete program class without exhaustive enumeration: a pretrained LLM induces a proposal distribution over candidate programs, each proposal is executed and scored on a held-out validation set, and the best program is selected, with no gradient updates or validation feedback used to adapt the sampling distribution. Across algorithmic tasks including parity variants, pattern matching, and primality testing, LLM-PV often recovers the exact underlying rule from a small labeled set and generalizes far beyond the training sequence lengths, while SGD-trained transformers, fine-tuning, in-context learning, and classical ML baselines can fit the training data yet fail to generalize reliably. Together, these results suggest that pretrained LLM priors can serve as effective search biases for ERM, narrowing the gap between statistical and computational efficiency.

## 1. Introduction

At its core, supervised learning asks for an algorithmic recipe for recovering structure from data: given labeled

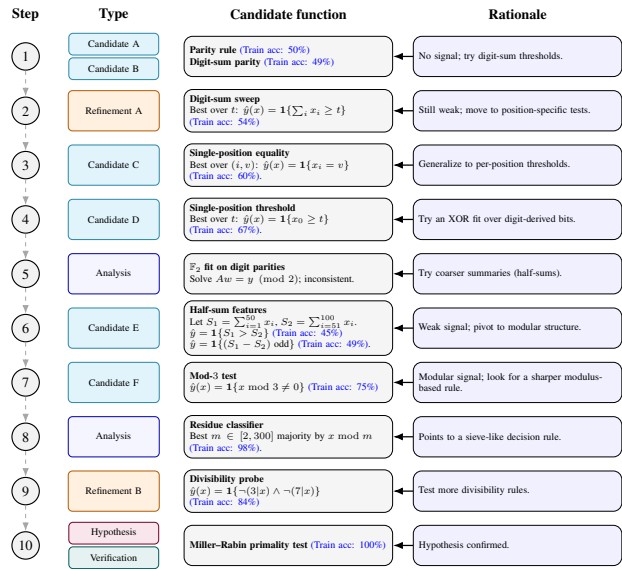

*Figure 1.* **Reasoning trace for learning primality.** The search starts from simple digit-level heuristics and progressively shifts toward modular structure: after parity and digit-sum baselines fail, residue-class predictors achieve high training accuracy, suggesting a sieve-like mechanism. A direct Miller–Rabin primality test then verifies the hypothesis and matches the labels perfectly.

examples $(x, y)$ from an unknown rule, produce a predictor that generalizes to unseen inputs. A central lesson of classical learning theory is that *simplicity* enables sample efficiency. In particular, for a finite hypothesis class $\mathcal{H}$, empirical risk minimization (ERM) needs only $\tilde{O}(\log |\mathcal{H}|)$ examples to generalize (Valiant, 1984; Vapnik, 1998). This immediately suggests a compelling perspective on *program learning*: if the target rule can be implemented by a short program of length $L$ over a token alphabet $\Sigma$, then the effective hypothesis class size is at most $|\Sigma|^L$, and hence $\tilde{O}(L \log |\Sigma|)$ labeled examples suffice.

The obstacle is computation. A direct implementation of ERM is length-first program enumeration: search all programs of length at most $L$ and return the first that fits the data. But the number of candidates grows exponentially, $|\mathcal{L}_{\leq L}| = \sum_{\ell=1}^{L} |\Sigma|^{\ell} = \Theta(|\Sigma|^L)$, so even when verification is fast, the overall runtime scales as $\Theta(m|\Sigma|^L)$. Thus, the classical route to sample-efficient learning of short programs is computationally infeasible for even moderate $L$

---

[1]Department of Computer Science & Engineering, Texas A&M University, TX, USA [2]Kempner Institute, Harvard University, MA, USA. Correspondence to: Tomer Galanti <galanti@tamu.edu>.

*Proceedings of the 43rd International Conference on Machine Learning*, Seoul, South Korea. PMLR 306, 2026. Copyright 2026 by the author(s).

(e.g., $L=20$ and $|\Sigma|=10$ already yields $10^{20}$ candidates).

**Modern deep learning flips this trade-off.** Instead of searching explicitly over programs, one typically trains high-capacity predictors by stochastic gradient descent (SGD) (Robbins & Monro, 1951; Bottou, 2010). This is computationally attractive: the optimization cost is often nearly linear in the data, and large models can fit complex training sets. Yet computational tractability does not imply statistical efficiency on structured rule families. In the statistical query (SQ) view (Kearns, 1998), broad classes of gradient-based procedures can require exponentially many samples on high–SQ-dimension targets such as parity- and cryptographic-like functions, despite the fact that these targets admit succinct program descriptions. In other words, the difficulty is not that the rule is long or unstructured, but that the learning algorithm accesses it through a bottlenecked interface (gradients, finite precision, local updates) that is poorly matched to discrete program structure.

> *Can we recover the sample efficiency of ERM over short programs without paying the full cost of exhaustive enumeration?*

**Contributions.** We focus on a small-data regime where the target has a short discrete description, so finite-class ERM enjoys strong classical generalization guarantees, yet the two standard computational routes are unsatisfying: length-first enumeration is exponential in program length, while gradient-based training can be sample-inefficient on structured, SQ-hard families. Our goal is not to introduce new PAC/SQ machinery or new constraint solvers. Instead, we make ERM over discrete program classes more computationally accessible by using pretrained LLMs only as a proposal mechanism for search, while keeping selection grounded in executable verification and held-out error. Concretely, we ask whether a pretrained LLM can serve as a useful *search prior* that reduces the need for exhaustive enumeration without changing the learning objective. Our contributions are:

- **A clean compute–sample tension for short programs.** Using classic SQ/PAC tools, we revisit a standard realizable family (planted $k$-parity) to formalize a regime that is central to small-data program learning. Finite-class ERM over length-$L$ programs achieves the usual sample bound $m = \tilde{O}(L \log |\Sigma|)$ (Eq. 1), but its runtime is exponential in $L$. In contrast, finite-precision mini-batch *coordinate* SGD may be computationally convenient per step, yet requires many fresh examples to reach nontrivial error on this family, namely $q = TB = \Omega(\sqrt{\binom{n}{k}}/2^b)$ (Prop. 3.1). Together, these results isolate a concrete small-data setting where both naive enumeration and gradient-based training break down, for complementary reasons.
- **ERM-style selection with an LLM proposal prior.** In-

spired by classic work on *programming by example* (PBE) and the growing literature on LLM-guided program synthesis, we introduce LLM-PV. LLM-PV is a propose-and-verify procedure that cleanly separates roles: (i) a pretrained LLM induces a data-dependent proposal distribution over candidate programs (or edits), (ii) an external verifier compiles and executes candidates, and (iii) we select by held-out validation error (Alg. 1). This makes the role of pretrained reasoning precise: it serves as a *search bias* that concentrates trials on plausible hypotheses, while the learning objective and selection rule remain similar to ERM.
- **Sample-efficient rule recovery and length generalization across algorithmic tasks.** Across algorithmic tasks including parity variants, pattern matching, palindromes, Dyck-2, primality variants and pseudo-random functions, LLM-PV often recovers the underlying rule from 200 labeled examples (Tabs. 1 and 2), and frequently outputs compact, input-length-invariant programs that generalize far beyond the training length (Fig. 3). In the same settings, classic ML methods (e.g., SVM, XGBoost), SGD-trained transformers and standard adaptation baselines (fine-tuning and in-context learning) commonly fit the training set but fail to generalize reliably as input length or dimension grows (Tab. 2, Fig. 4).
- **Auditability of both the learned hypothesis and the search process.** The output is executable, human-readable code, and the full learning trajectory is inspectable: we log the sequence of proposed programs/edits, execution outcomes, and verification diagnostics, yielding an auditable propose–verify trace (Fig. 1 and Fig. 7 in App. A.4). This enables direct debugging of failure modes and mechanistic validation of successes, beyond reporting final accuracy.

## 2. Related Work

**PAC learning, Occam's razor and short programs.** We follow the the classical generalization theory, where finite-class ERM has sample complexity $O(\log |\mathcal{H}|)$ (Valiant, 1984; Vapnik & Chervonenkis, 1971; Vapnik, 1998). The "short program" view instantiates Occam/MDL: a hypothesis encodable in $L$ symbols over alphabet $\Sigma$ admits bounds of order $O(L \log |\Sigma|)$, up to confidence terms (Blumer et al., 1987; Barron & Cover, 1991; Barron et al., 1998; Rissanen, 1989; McAllester, 1998). The length-first search (Alg. 2) realizes this ERM guarantee but incurs exponential time in description length, reflecting the classic universal-search trade-off (Levin, 1973; Solomonoff, 1964a;b).

**Statistical query (SQ) learning and hardness of learning.** The SQ framework and its refinements (Kearns, 1998; Blum et al., 1994; Feldman, 2017; Reyzin, 2020) yield lower bounds for many concept classes. Parity and related families

have large SQ dimension under the uniform distribution, so any SQ learner needs exponentially many (tolerant) queries to achieve nontrivial correlation (Blum et al., 1994; Feldman et al., 2017; Klivans & Sherstov, 2007; Klivans & Kothari, 2014; Giapitzakis et al., 2025). Intuitively, mini-batch SGD is itself approximately an SQ algorithm: each update averages a bounded statistic over samples (Feldman et al., 2017; 2018; Abbe et al., 2021; Barak et al., 2022). Hence, SQ lower bounds transfer directly to SGD, making its iteration complexity grow with the SQ dimension—exponentially for parities and pseudorandom families under the uniform distribution. Our analysis formalizes this connection, showing how SQ hardness induces exponential sample requirements for gradient-based methods.

**Gradient-based training on algorithmic reasoning.** This SQ perspective aligns with extensive empirical evidence: even when a neural family can represent the target compactly, SGD-trained networks often fail on parity-like and compositional algorithmic tasks without strong inductive bias or large data (Shalev-Shwartz et al., 2017; Safran & Shamir, 2018; Daniely & Malach, 2020; Barak et al., 2022). More broadly, work on neural *trainability* separates representational power from optimization and sample efficiency (Yehudai & Shamir, 2019; Daniely, 2017), and phenomena such as "grokking"—delayed generalization after long training—highlight a gap between statistical optima and what SGD finds in practice (Power et al., 2022). These observations motivate alternatives that retain finite-class guarantees while improving practical search efficiency.

**LLM-guided search and iterative improvement.** Program synthesis and discrete search over programs has been studied for decades, and it has long been viewed as a difficult problem because the hypothesis space is combinatorial and brittle to small specification changes. The LLM era introduced a new primitive: strong *language-conditioned priors* over discrete objects, which can be used as proposals in a search loop. A prominent thread uses iterative propose–critique–revise driven by natural-language feedback (e.g., Self-Refine, Reflexion), and related approaches treat feedback as an optimization signal, including so-called textual gradients (Madaan et al., 2023; Shinn et al., 2023; Yuksekgonul et al., 2024). In parallel, evolutionary and neuro-symbolic frameworks use LLMs to propose edits or modular building blocks that are refined via mutation and selection (Pourcel et al., 2025; Novikov et al., 2025; DeepMind, 2025; Bhansali et al., 2024).

**Programming by examples and LLM-assisted program synthesis.** A complementary tradition studies *programming by examples* (PBE) and inductive program synthesis, where the input is a finite set of input–output examples or tests and the goal is to output a program consistent with them (Gulwani, 2011; Le & Gulwani, 2014; Singh & Gulwani,

2012; Alur et al., 2013; Gulwani et al., 2017). This literature emphasizes search over discrete program spaces (often within a DSL), including enumerative synthesis, version-space algebras, and CEGIS-style loops (Lau et al., 2003; Solar-Lezama et al., 2006; Polikarpova et al., 2016; Solar-Lezama, 2013), often complemented by ranking, inductive bias, or user interaction to resolve ambiguity (Gulwani, 2011; Gulwani et al., 2017; Alur et al., 2013).

Recent work revisits PBE in the LLM era. One early paradigm treats the LLM itself as the "program" via *in-context learning* (ICL): demonstrations, often with chain-of-thought, induce task procedures without parameter updates (Brown et al., 2020; Min et al., 2022; Wei et al., 2022), and this phenomenon has been analyzed theoretically (Von Oswald et al., 2023; Akyürek et al., 2023; Shen et al., 2024; de Wynter, 2025). Empirically, however, pretrained LLMs can be brittle when asked to synthesize *explicit* programs from examples alone, with performance depending strongly on distributional match and fine-tuning (Li & Ellis, 2024). Other work proposes LLM-guided synthesis pipelines that retain execution-based verification while improving robustness via decomposition and compositional search (Khan et al., 2025). Complementary benchmarks study *interactive* settings, where an agent can query a hidden target function and refine solutions using feedback or counterexamples (Wei et al., 2025). More broadly, LLMs have been combined with program analysis and synthesis components to improve the reliability of generated code in practical settings (Jain et al., 2022).

**Our lens.** We adopt the propose–verify view from program synthesis and PBE and repurpose it for *learning from i.i.d. data* with an explicit generalization objective. We treat the hypothesis space as a discrete class of executable programs and use a pretrained LLM only as an inductive-bias mechanism: a proposal distribution over candidate programs. Each proposal is verified by execution on the observed samples, and the final program is chosen by validation-based model selection, so the criterion is held-out error rather than mere consistency with a finite specification. In this sense, we inherit the verification discipline of synthesis, but replace interactive counterexamples and hand-crafted templates with distributional learning and ERM-style selection from $S$.

## 3. Theoretical Analysis

### 3.1. Problem Setup

We study *inductive program synthesis* ("program learning"): the target is a binary function $y : \mathcal{X} \to \{\pm 1\}$ implemented by a short program in a fixed language, and the learner receives i.i.d. examples $S = \{(x_i, y(x_i))\}_{i=1}^m$ with $x_i \overset{\text{i.i.d.}}{\sim} D$. Throughout, we assume the *realizable* setting, i.e., $y \in \mathcal{L}$, where $\mathcal{L}$ is the class of total functions computed by

programs in the language (formalized below).

**Language and semantics.** Fix a finite alphabet $\Sigma$ and a programming language $\mathcal{L} \subseteq \Sigma^*$. Each string $u \in \mathcal{L}$ has semantics $[\![u]\!] : \mathcal{X} \rightharpoonup \{\pm 1\}$, a (possibly partial) function that may fail to compile or fail to halt. We write $[\![u]\!](x) = \bot$ when $u$ does not produce an output on $x$. Let $\mathcal{C} := \{ f : \mathcal{X} \rightarrow \{\pm 1\} : \exists u \in \mathcal{L} \text{ s.t. } [\![u]\!] \text{ is total and } [\![u]\!] = f \}$. We denote the length of $u$ by $|u|$ (in symbols over $\Sigma$) and write $\mathcal{L}_\ell := \{u \in \mathcal{L} : |u| = \ell\}$. A program is considered total if it defines an output for every input—i.e., it never fails to compile and halts on all $x \in \mathcal{X}$, returning a $\pm 1$ label.

**Objective.** For a hypothesis $h : \mathcal{X} \rightarrow \{\pm 1\}$, define population error $\mathrm{err}_D(h) := \Pr_{x \sim D}[h(x) \neq y(x)]$ and empirical error $\mathrm{err}_S(h) := \frac{1}{m} \sum_{i=1}^m \mathbf{1}\{h(x_i) \neq y_i\}$. The goal is to output a program $u \in \mathcal{L}$ whose total semantics $[\![u]\!]$ attains small $\mathrm{err}_D$.

**Computational model.** When executing a candidate program $u$ on input $x$, we allow a time budget $T \in \mathbb{N}$ per call; if $u$ fails to compile or does not halt within time $T$, we treat the outcome as $\bot$ and reject $u$ as a hypothesis. This makes search procedures well-defined even when $[\![u]\!]$ is partial.

**Short-program regime.** We will frequently analyze the *short-program* subclass $\mathcal{H}_\ell := \{[\![u]\!] : u \in \mathcal{L}_\ell, [\![u]\!] \text{ total}\}$, where $\mathcal{H} = \bigcup_{\ell \geq 1} \mathcal{H}_\ell$ and compare (i) explicit search over $\mathcal{H}_\ell$ (finite-class ERM) to (ii) gradient-based learners $h_\theta$ drawn from a proxy hypothesis family $\{h_\theta : \theta \in \Theta\}$.

### 3.2. Program Enumeration

One approach to program learning is *program enumeration*. Given a language $\mathcal{L}$ with token alphabet $\Sigma$, enumerate candidate programs in a canonical order (e.g., by length and then lexicographically), and return the first program that is consistent with the training sample (Alg. 2 in App. B.1).

Despite its simplicity, enumeration is well known to be highly sample efficient. By standard PAC bounds for finite hypothesis classes (Valiant, 1984) (see also Cor. 2.3 of (Shalev-Shwartz & Ben-David, 2014)), if the target function can be implemented by some length-$L$ program in $\mathcal{L}$, then with probability at least $1 - \delta$ over the selection of $S$, the returned program $h$ satisfies

$$\mathrm{err}_D(h) \leq m^{-1}[L \log |\Sigma| + \log(2L^2/\delta)]. \quad (1)$$

Thus, the required sample size scales with the program length of the target function. For completeness, we prove this inequality in App. B.1.

The main limitation is computational. In the worst case, length-first enumeration must examine essentially all programs up to length $L$, which is exponential in $L$: $\Omega(|\Sigma|^L)$. For example, with an ASCII-sized alphabet $|\Sigma| = 128$ and $L = 10$, this corresponds to $128^{10} \approx 1.2 \times 10^{21}$ candidate

strings, making brute-force search infeasible in practice.

### 3.3. Gradient-Based Optimization

A alternative method is to search for a predictor in a large parametric class (e.g., neural networks) using gradient-based optimization. To reason about the information such methods obtain from data, we use the statistical-oracles viewpoint: many iterative optimization procedures can be modeled as algorithms that access the data distribution only through estimates of expectations of bounded functions and closely related sample-based variants (Kearns, 1998; Feldman et al., 2018; 2017; Reyzin, 2020).

**Finite-precision SGD.** There are two standard ways to connect SGD to statistical-oracles: either inject independent noise so each update becomes a tolerant expectation query, or focus on finite-precision coordinate mini-batch SGD where each per-example coordinate contribution simulates a 1-STAT($b$) query and $q = TB$ such queries can be simulated via VSTAT by the standard reduction (Feldman et al., 2018, Thm. B.4). We therefore model mini-batch coordinate SGD as an algorithm that only accesses fresh data through *finite-precision* (per-example) gradient information. At iteration $t$, the algorithm selects parameters $\theta_t$ and a coordinate $j_t \in [d]$, draws a fresh mini-batch $z_{t,1}, \ldots, z_{t,B} \sim D$, and updates using the $b$-bit quantized mini-batch coordinate gradient $\theta_{t+1} = \theta_t - \eta_t \widehat{g}_{t,j_t}^{(b)} e_{j_t}$, $\widehat{g}_{t,j_t}^{(b)} := \frac{1}{B} \sum_{i=1}^B Q_b (\partial_{j_t} \ell(\theta_t, z_{t,i}))$, where $Q_b$ is a $b$-bit quantizer and $\eta_t$ is the step size. The only interaction with fresh samples is via these $b$-bit per-example gradient values.

**Planted $k$-parity.** A canonical SQ-hard family is parity under the uniform distribution, and closely related planted parity testing problems inherit the same "low-correlation" structure used in SQ lower bounds (Blum et al., 2003; Kearns, 1998; Reyzin, 2020; Feldman et al., 2017). Fix $n \geq 1$ and $k \in \{1, \ldots, n\}$, let $\mathcal{X} = \{0, 1\}^n$, and let $\mathcal{S}_k$ be the set of $k$-sparse vectors in $\{0, 1\}^n$. For each $s \in \mathcal{S}_k$, define the target $f_s(x) = (-1)^{\langle s, x \rangle}$ and the realizable distribution $D_s$ on $\mathcal{X} \times \{\pm 1\}$ by $x \sim \mathrm{Unif}(\mathcal{X})$ and $y = f_s(x)$. Let $N := |\mathcal{S}_k| = \binom{n}{k}$ be the number of possible targets.

The following is a technical application of using the two-step reduction using (Feldman et al., 2018, Thm. B.4) and the classic results from SQ learning (Reyzin, 2020). For completeness, we provide a full formal analysis in App. C.

**Proposition 3.1.** *Consider any (possibly adaptive, randomized) procedure that runs $T$ iterations of $b$-bit mini-batch coordinate SGD with batch size $B$ on the above family, meaning that on fresh examples $(x, y) \sim D_s$ its only access to the data is through $b$-bit quantized per-example coordinate gradients as in the update displayed above. Let the procedure output a hypothesis $h : \mathcal{X} \rightarrow \{\pm 1\}$, and write $q := TB$ for the total number of fresh examples used.*

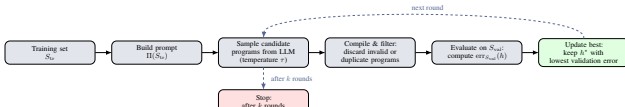

*Figure 2.* **An illustration of LLM-PV.** A prompt built from $S_{\mathrm{tr}}$ conditions the LLM to sample candidate programs. We compile and filter candidates, evaluate validation error on $S_{\mathrm{val}}$, and maintain the best-so-far hypothesis $h^\star$. The procedure runs for $k$ rounds and returns the program with the lowest validation error among all candidates evaluated.

*If for every $s \in \mathcal{S}_k$ the procedure achieves nontrivial population error $\mathrm{err}_{D_s}(h) \leq \frac{1}{4}$ with probability at least $\beta \geq 5/6$, then necessarily $q = \Omega\left(\frac{\sqrt{N}}{2^b}\right)$.*

Program enumeration (Alg. 2) and finite-precision coordinate SGD fail for planted $k$-parity for complementary reasons. Let $N = \binom{n}{k}$ be the number of $k$-parities. Enumerating programs of length $\leq L$ costs $\Theta(m, |\Sigma|^L)$. A $k$-parity has a description of length $L = \Theta(\log N)$ (ASCII/Python), and Eq. 1 gives $m = \tilde{O}(L) = \tilde{O}(\log N)$ for nontrivial error, so the resulting work is $\Theta(|\Sigma|^{\Theta(\log N)} \log N) = N^{\Theta(1)} \log N$, i.e., polynomial in $N$. By contrast, finite-precision coordinate SGD runs in $\Theta(m \cdot \mathrm{Cost}\nabla)$ time on $m$ fresh examples, but Prop. 3.1 needs $m = \Omega(\sqrt{N}/2^b)$ to reach population error $\leq 1/4$, yielding total work $\Omega((\sqrt{N}/2^b) \cdot \mathrm{Cost}\nabla)$. In short: enumeration is expensive in *search* (about $N^{\Theta(1)}$ candidates), while SGD is expensive in *data* (about $\sqrt{N}$ fresh examples), and both become infeasible as $N = \binom{n}{k}$ grows (for constant $k$, $N = \Theta(n^k)$ so costs scale as $n^{\Theta(k)}$ vs. $n^{k/2}$, up to $\mathrm{Cost}_\nabla$).

## 4. New Lens

A natural alternative to brute-force program enumeration is *sampling with verification*: draw candidates and return the one with smallest held-out error. But *uniform* sampling is hopeless: if $u \sim \mathrm{Unif}(\mathcal{L}_L)$ and the target is $u^\star$, then $\Pr[u = u^\star] = |\Sigma|^{-L}$, so the expected trials are $|\Sigma|^L$, essentially matching enumeration. Thus the bottleneck is not verification, but the *proposal distribution*.

This motivates a *proposer* that, given $S_{\mathrm{tr}}$, induces a data-dependent distribution $q(\cdot \mid S_{\mathrm{tr}})$ placing non-negligible mass on programs consistent with the observed pairs. Given such a proposal, the rest is standard: compile/execute candidates, discard invalid programs, and select by validation error.

A strong pretrained LLM provides a natural proposal bias: prompted with $S_{\mathrm{tr}}$, it generates code concentrated on recognizable templates and concise rules suggested by the examples, rather than spreading mass nearly uniformly over the vast program space. In this view, the LLM is used as a data-dependent *proposal prior distribution*.

**Algorithm 1** LLM-PV: $k$-try LLM propose-and-verify with validation

**Require:** train/validation sets $S_{\mathrm{tr}}, S_{\mathrm{val}}$; trials $k$; prompt builder $\Pi$; temperature $\tau$
**Ensure:** Best program $u^\star$ with $h^\star = [\![u^\star]\!]$
  1: Initialize $\mathrm{err}^\star \leftarrow 1$, $(u^\star, h^\star) \leftarrow (\bot, \bot)$, $\mathcal{U} \leftarrow \emptyset$
  2: **for** $t = 1, \ldots, k$ **do**
  3:     Sample a candidate program $u$ from $\mathrm{LLM}(\Pi(S_{\mathrm{tr}}); \tau)$
  4:     **if** $u \in \mathcal{U}$ **then**
  5:         **continue**
  6:     **end if**
  7:     $\mathcal{U} \leftarrow \mathcal{U} \cup \{u\}$; compile and set $h \leftarrow [\![u]\!]$
  8:     **if** $h$ undefined on some $x \in S_{\mathrm{tr}} \cup S_{\mathrm{val}}$ **then**
  9:         **continue**
 10:     **end if**
 11:     Compute $\mathrm{err}_{S_{\mathrm{val}}}(h)$
 12:     **if** $\mathrm{err}_{S_{\mathrm{val}}}(h) < \mathrm{err}^\star$ **then**
 13:         $(\mathrm{err}^\star, u^\star, h^\star) \leftarrow (\mathrm{err}_{S_{\mathrm{val}}}(h), u, h)$
 14:     **end if**
 15: **end for**
 16: **return** $(u^\star, h^\star)$

Alg. 1 instantiates this propose–verify–select pipeline. Based on Thm. 11.1 in (Shalev-Shwartz & Ben-David, 2014), the guarantee depends on the *proposal mass* $p_\epsilon(S_{\mathrm{tr}}) := \Pr_{h \sim q(\cdot \mid S_{\mathrm{tr}})}[\mathrm{err}_D(h) \leq \epsilon]$, plus the usual finite-class validation penalty.

**Proposition 4.1.** *Fix $\epsilon \geq 0$ and $\delta \in (0,1)$. Draw independent samples $S_{\mathrm{tr}} \sim D^{m_{\mathrm{tr}}}$ and $S_{\mathrm{val}} \sim D^{m_{\mathrm{val}}}$ with labels from the target $y : \mathcal{X} \to \{\pm 1\}$. Run Alg. 1 for $k$ trials, and assume each trial outputs an accepted total hypothesis $h_t : \mathcal{X} \to \{\pm 1\}$ using only $S_{\mathrm{tr}}$. Let $h^\star \in \arg\min_{t \in [k]} \mathrm{err}_{S_{\mathrm{val}}}(h_t)$ be the returned hypothesis. If $k \geq \lceil \frac{\log(\delta/2)}{\log(1 - p_\epsilon(S_{\mathrm{tr}}))} \rceil$, then with probability at least $1 - \delta$ over $(S_{\mathrm{tr}}, S_{\mathrm{val}})$ (and all algorithmic randomness), $\mathrm{err}_D(h^\star) \leq \epsilon + 2\sqrt{\log(4k/\delta)/(2m_{\mathrm{val}})}$.*

The guarantee separates the two roles in LLM-PV. The LLM enters through $p_\epsilon(S_{\mathrm{tr}})$, the probability that a fresh accepted proposal has population error at most $\epsilon$. Given $k$ candidates, validation selection pays the standard finite-class price $\tilde{O}(\sqrt{\log(k/\delta)/m_{\mathrm{val}}})$. Thus the LLM need not be trusted as a predictor; it only needs to put enough mass on low-error programs so one appears within $k$ trials, after which performance is certified by held-out evaluation.

**Comparing the number of programs observed.** Uniform sampling from $\mathcal{L}_L$ hits any fixed program with probability $|\Sigma|^{-L}$, so it takes on the order of $|\Sigma|^L$ trials to observe it. In LLM-PV the relevant quantity is the proposal mass $p_\epsilon(S_{\mathrm{tr}})$ on $\epsilon$-good hypotheses: by Prop. 4.1 it suffices that $k \geq \lceil \frac{\log(2/\delta)}{-\log(1 - p_\epsilon(S_{\mathrm{tr}}))} \rceil$, and for $p_\epsilon(S_{\mathrm{tr}}) \ll 1$ this

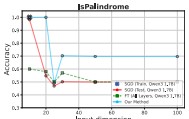 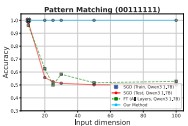 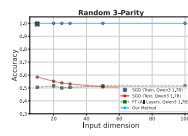

*Figure 3.* **LLM-PV generalizes to much larger input lengths.** All methods (SGD, FT, and LLM-PV) are trained on 200 examples at a fixed length ($n=10$) and evaluated on lengths up to $n=100$. SGD learns the training distribution but degrades to near-chance accuracy ($\approx 50\%$) as $n$ grows. In contrast, LLM-PV maintains high accuracy and, on Pattern Matching and Random 3-Parity, synthesizes dimension-invariant programs.

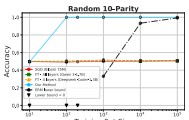 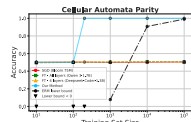 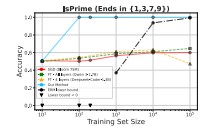

*Figure 4.* **SGD-trained LLMs fail to learn algorithmic tasks even with 100k examples; LLM-PV succeeds with 200.** We compare the test performance of LLM-PV and SGD training (from-scratch and fine-tuning) on 10-P **(left)**, CAP **(middle)**, and IsPrime **(right)** when varying the training set size. **SGD training overfits, with test accuracy near chance across all training sizes. LLM-PV achieves 100% test accuracy with just 200 examples.**

is $k \approx \frac{\log(2/\delta)}{p_\epsilon(S_{\mathrm{tr}})}$. Thus, LLM-PV replaces the exponential search size $|\Sigma|^L$ by an *effective* search size $1/p_\epsilon(S_{\mathrm{tr}})$, improving over enumeration whenever $1/p_\epsilon(S_{\mathrm{tr}}) \ll |\Sigma|^L$ (equivalently $p_\epsilon(S_{\mathrm{tr}}) \gg |\Sigma|^{-L}$).

**Interpretability.** LLM-PV makes both the *learned object* and the *learning process* transparent: each run returns (i) an executable program and (ii) a reasoning trace that logs the proposed candidates and their observed errors (often including train accuracy). Fig. 1 shows a `GPT-5-Thinking` trace (ChatGPT web UI) on IsPrime with $m=100$ examples of 100-digit inputs: it starts with simple digit-level rules (parity and digit-sum tests), then shifts to modular structure, and ends with a Miller–Rabin primality test. Because each step is linked to the errors it aims to fix, failures are auditable and successes are inspectable: the final symbolic program can be unit-tested, stress-tested out of distribution, and re-run or edited.

# 5. Experiments

## 5.1. Setup

We evaluate whether LLM-guided propose-and-verify can recover short executable rules from few labeled examples, and compare it to standard learning baselines. Each task is binary classification with inputs $x \in \mathcal{X}$ (bitstrings/digit strings/graph encodings) and labels $y(x) \in \{0, 1\}$. We draw $m=200$ i.i.d. labeled examples and split them into $S_{\mathrm{tr}}$ and $S_{\mathrm{val}}$ of size 100 each. We report test accuracy on an independent held-out set of 10,000 examples.

### 5.1.1. EVALUATION TASKS

**Synthetic tasks.** We use synthetic algorithmic tasks spanning non-local properties and simple heuristics: **(1)** parity variants: full parity (FP), first-half parity (FHP), random 3-parity (3-P), random 10-parity (10-P); **(2)** pattern matching for fixed motifs `00111111` (Pattern1) and `10101010` (Pattern2); **(3)** palindrome detection (IsPal); **(4)** Dyck-2 membership (Dyck-2); **(5)** primality (IsPrime) while restricting negatives to last digits $\in \{1, 3, 7, 9\}$; **(6)** the shifted-input primality variant (IsPrime+47), where we add 47 to each integer before labeling; **(7)** cellular automata parity (CAP); and **(8)** SHA-256 parity (SHA). All datasets are class-balanced by construction. Formal definitions and data generation appear in App. A.1.

Despite being familiar benchmarks, these targets are difficult to learn from i.i.d. labeled examples without an explicit algorithmic bias. First, $k$-parity suffers from a combinatorial search barrier: the label depends on a tiny unknown subset of coordinates, and there are $\binom{100}{10}$ candidate 10-bit subsets over 100 variables, so fitting low-order correlations provides essentially no guidance toward the correct rule. Second, palindromes and Dyck-2 have hard negatives that differ from positives by only a few edits, which preserves most local $n$-gram statistics; thus, any learner that relies primarily on local cues will struggle to separate the classes. For Dyck-2, we additionally encode the two bracket types as a 2-bit alphabet $\{00, 01, 10, 11\}$ to remove trivial formatting cues; success must come from tracking the underlying stack-like constraint rather than from recognizable characters. Third, motif detection is a needle-in-a-haystack problem: a short pattern can occur at an unknown position within a long string, so generalization requires reasoning that is robust to shifts rather than exploiting fixed coordinates. Fourth, cellular-automata parity and SHA-based labels exhibit an avalanche effect, where small input changes can induce global output changes, ruling out simple heuristic rules based on a few digits or local patterns. Finally, for 100-digit primality, memorization is implausible given the astronomically large number of candidates (on the order of $10^{97}$ primes), so systematic success requires implementing an algorithmic test; moreover, IsPrime+47 makes this necessity explicit by defining the label via the primality of $\mathrm{int}(x) + 47$, so any strategy that treats the label as a direct property of the observed digit string $x$ will fail.

## 5.2. Methods Compared

We compare LLM-PV to several families of representative baselines. To keep the main text focused, we describe each baseline at a high level here and defer full hyperparameters, prompts, and sweeps to App. A.2.

**LLM-PV.** We instantiate Alg. 1 with several backbone models, including open-source

| Model / Task | FP | FHP | Pattern1 | Pattern2 | 3-P | 10-P | IsPal | Dyck-2 | CAP | IsPrime | IsPrime+47 | CycleCount | SHA |
|---|---|---|---|---|---|---|---|---|---|---|---|---|---|
| XGBoost | 49.9 | 50.0 | 50.6 | 52.3 | 49.9 | 49.3 | 83.6 | 49.7 | 50.4 | 61.8 | 59.3 | 51.1 | 50.3 |
| Random Forest | 49.8 | 49.7 | 50.5 | 52.5 | 49.7 | 49.3 | 77.8 | 50.1 | 49.2 | 58.7 | 55.7 | 50.8 | 50.6 |
| SVM | 49.9 | 49.9 | 49.8 | 51.6 | 50.6 | 48.9 | 69.7 | 50.1 | 50.1 | 55.6 | 55.7 | 47.0 | 50.2 |
| Genetic Algorithm | 49.8 | 52.5 | 51.1 | 49.7 | 47.0 | 54.0 | 87.1 | 51.9 | 48.6 | 51.7 | 53.5 | 49.3 | 49.0 |
| TabPFN (Hollmann et al., 2022) | 50.1 | 50.5 | 49.2 | 51.2 | 49.3 | 50.5 | 49.4 | 50.5 | 48.0 | 57.1 | 65.4 | 51.3 | 50.6 |
| SGD: Qwen3-1.7B | 49.3 | 50.6 | 51.5 | 53.6 | 49.7 | 50.3 | 49.2 | 51.4 | 49.4 | 58.8 | 53.6 | 53.5 | 49.8 |
| SGD: DeepSeek-Coder-1.3B | 52.2 | 50.5 | 51.3 | 55.1 | 54.3 | 51.3 | 51.6 | 49.4 | 50.0 | 54.5 | 56.0 | 53.9 | 49.8 |
| SGD: Llama-3.2-1B | 52.2 | 48.6 | 50.0 | 54.3 | 51.5 | 50.6 | 50.3 | 51.5 | 50.2 | 52.7 | 51.7 | 54.7 | 51.0 |
| FT: Llama-3.2-1B | 50.4 | 51.3 | 61.5 | 57.6 | 51.0 | 50.5 | 49.6 | 52.7 | 50.7 | 57.2 | 58.7 | 65.7 | 50.8 |
| FT: Qwen3-1.7B | 51.2 | 50.7 | 59.6 | 59.4 | 50.7 | 50.4 | 49.6 | 52.5 | 50.8 | 58.3 | 52.9 | 57.4 | 51.2 |
| FT: DeepSeek-Coder-1.3B | 50.5 | 50.2 | 56.4 | 57.7 | 50.7 | 50.4 | 49.7 | 51.7 | 50.2 | 58.9 | 60.3 | 55.7 | 49.7 |
| ICL: Qwen3-30B-Instruct (no tools) | 53.0 | 54.0 | 56.0 | 51.0 | 50.0 | 50.0 | 47.0 | 48.0 | 49.0 | 50.0 | 58.0 | 46.0 | **54.0** |
| ICL: Qwen3-Coder-30B-Instruct (no tools) | 50.0 | 50.0 | 50.0 | 50.0 | 50.0 | 44.0 | 51.0 | 50.0 | 49.0 | 47.0 | 60.0 | 47.0 | 49.0 |
| ICL: DeepSeek-Coder-33B-Instruct (no tools) | 43.0 | 54.0 | 54.0 | 47.0 | 47.0 | 48.0 | 51.0 | 54.0 | 44.0 | 53.0 | 61.0 | 41.0 | 52.0 |
| ICL: Gemini-3.1-Pro (with tools) | 48.0 | 54.0 | 83.0 | 57.0 | 50.0 | 45.0 | **100** | 51.0 | 51.0 | 52.0 | 56.0 | 50.0 | 45.0 |
| ICL: GPT-5-Thinking (no tools) | 82.0 | 66.0 | 55.0 | 41.0 | 47.0 | 47.0 | 79.0 | 57.0 | 51.0 | 55.0 | 47.0 | 43.0 | 44.0 |
| ICL: GPT-5-Thinking (with tools) | 99.0 | 88.0 | 91.0 | 51.0 | 75.0 | 79.0 | 91.0 | 59.0 | 46.0 | 62.0 | 56.0 | 51.0 | 44.0 |
| MLAgentBench (Huang et al., 2024): GPT-5-Thinking (with tools) | 53.6 | 51.0 | 33.4 | 60.9 | 32.8 | 50.1 | 31.6 | 52.1 | 50.0 | 32.6 | 33.1 | 49.1 | 49.9 |
| AIDE-ML (Jiang et al., 2025): GPT-5-Thinking (with tools) | 49.5 | 49.9 | 50.3 | 50.9 | 49.5 | 49.2 | 50.0 | 50.5 | 50.2 | 55.5 | 49.7 | 50.0 | 50.8 |
| LLM-PV: Qwen3-30B-Instruct (no tools) | 50.4 | 49.6 | 51.1 | 60.9 | 50.4 | 51.6 | 49.9 | 80.4 | 50.2 | 50.6 | 50.2 | 49.9 | 49.7 |
| LLM-PV: DeepSeek-Coder-33B-Instruct (no tools) | **100** | 50.9 | 50.9 | 50.5 | 51.6 | 50.8 | 51.0 | 57.3 | 50.2 | 50.0 | 50.2 | 50.0 | 51.5 |
| LLM-PV: OLMo-3-32B-Instruct (no tools) | **100** | 50.5 | 50.0 | 83.5 | 50.2 | 50.2 | 52.7 | 50.5 | 50.8 | 50.8 | 50.5 | 52.7 | 50.1 |
| LLM-PV: Gemini-3.1-Pro (with tools) | **100** | **100** | **100** | **100** | **100** | **100** | **100** | 80.7 | 52.9 | **100** | 68.1 | 96.1 | 50.3 |
| LLM-PV: GPT-5-Thinking (no tools) | **100** | 51.1 | 78.4 | 64.7 | 50.1 | 50.4 | 50.0 | 79.9 | 50.2 | 52.9 | 64.0 | 51.0 | 50.2 |
| LLM-PV: GPT-5-Thinking (with tools) | **100** | **100** | **100** | **100** | **100** | **100** | **100** | **100** | **100** | **100** | **80.6** | **96.9** | 50.1 |

Table 1. **Test accuracy at length** $n{=}100$ **across baselines and LLM-PV.** We compare classic ML algorithms, training from scratch (SGD on 1B-scale LMs), fine-tuning (top-$k$ layers or full) of 1B-scale pretrained LMs, and in-context learning with 30B+ instruction-tuned models. **Most baselines remain near chance on the algorithmic tasks at $n{=}100$, while LLM-PV achieves high test accuracy on all tasks except SHA.**

| | SGD | | | | | FT | | | | | ICL | | | | | LLM-PV | | | | |
|---|---|---|---|---|---|---|---|---|---|---|---|---|---|---|---|---|---|---|---|---|
| Task | $n{=}20$ | $n{=}25$ | $n{=}30$ | $n{=}50$ | $n{=}100$ | $n{=}20$ | $n{=}25$ | $n{=}30$ | $n{=}50$ | $n{=}100$ | $n{=}20$ | $n{=}25$ | $n{=}30$ | $n{=}50$ | $n{=}100$ | $n{=}20$ | $n{=}25$ | $n{=}30$ | $n{=}50$ | $n{=}100$ |
| FP | 56.3 | 56.2 | 52.4 | 52.7 | 52.2 | 50.8 | 51.0 | 50.4 | 51.2 | 51.2 | 52.0 | 47.0 | 54.0 | 51.0 | 53.0 | **100**[†] | **100**[†] | **100**[†] | **100**[†] | **100**[†] |
| FHP | 51.0 | 52.5 | 50.7 | 53.7 | 50.6 | 52.7 | 51.7 | 50.8 | 50.5 | 51.3 | 50.0 | 50.0 | 46.0 | 59.0 | 54.0 | **100** | **100** | **100** | **100** | **100** |
| 3-P | 50.6 | 50.5 | 52.2 | 51.4 | 54.3 | 50.9 | 50.7 | 53.3 | 51.0 | 51.0 | 54.0 | 56.0 | 51.0 | 48.0 | 50.0 | **100** | **100** | **100** | **100** | **100** |
| 10-P | 50.6 | 50.2 | 50.3 | 50.6 | 50.3 | 51.1 | 50.5 | 50.8 | 50.3 | 50.7 | 51.0 | 55.0 | 55.0 | 51.0 | 50.0 | **100** | **100** | **100** | **100** | **100** |
| Pattern1 | 91.4 | 82.8 | 57.8 | 58.7 | 51.5 | 74.6 | 72.0 | 68.4 | 68.5 | 61.5 | 89.0 | 69.0 | 77.0 | 61.0 | 56.0 | **100**[†] | 98.9 | 98.5 | **100**[†] | **100**[†] |
| Pattern2 | 97.5 | **96.0** | **95.2** | 74.5 | 55.1 | 69.4 | 69.2 | 67.4 | 67.3 | 59.4 | 79.0 | 68.0 | 72.0 | 79.0 | 51.0 | **100**[†] | 94.2 | 93.2 | **100**[†] | **100**[†] |
| IsPal | 62.7 | 56.5 | 54.8 | 51.6 | 51.6 | 52.4 | 51.3 | 51.1 | 50.2 | 49.7 | 58.0 | 53.0 | 63.0 | 52.0 | 51.0 | **100** | 96.0 | **100** | **100** | **100** |
| Dyck-2* | 65.7 | 59.1 | 55.9 | 51.1 | 51.8 | 64.6 | 57.9 | 53.9 | 52.8 | 52.7 | 70.0 | 57.0 | 61.0 | 54.0 | 54.0 | **77.4** | **90.5** | **80.0** | **90.5** | **100** |
| IsPrime | 59.9 | 60.2 | 63.0 | 57.0 | 58.8 | 59.2 | 59.8 | 59.2 | 62.5 | 58.9 | 53.0 | 53.0 | 53.0 | 57.0 | 53.0 | **100**[†] | **100**[†] | **100**[†] | **100**[†] | **100**[†] |
| CAP◇ | 52.5 | 51.4 | 52.0 | 50.5 | 50.0 | 50.9 | 50.6 | 51.2 | 51.1 | 50.8 | 52.0 | 47.0 | 46.0 | 50.0 | 49.0 | **100** | **100** | **100**[†] | **100**[†] | **100**[†] |

[†] indicates 100% for all $n$. *For Dyck-2, lengths are $n \in \{20, 40, 60, 80, 100\}$ respectively.

Table 2. **LLM-PV vs. training and prompting baselines.** Test accuracy (%) for input lengths $n \in \{20, 25, 30, 50, 100\}$. **SGD**: best test accuracy across Qwen3-1.7B, Deepseek-Coder-1.3B, and Llama3.2-1B, each trained from scratch. **FT**: best test accuracy across Llama3.2-1B, Qwen3-1.7B, and Deepseek-Coder-1.3B, sweeping tuned layers ({top-2, top-4, top-8, full}) and training hyperparameters, separately for each (task,$n$). **ICL**: best test accuracy across Qwen3-30B-Instruct, Qwen3-Coder-30B-Instruct, and Deepseek-Coder-33B-Instruct.

models such as Qwen3-30B-Instruct, Deepseek-Coder-33B-Instruct, and Olmo-3-32B-Instruct, as well as closed models such as Gemini-3.1-Pro and GPT-5-Thinking. The open-source models are evaluated without tool calls. For GPT-5-Thinking, we explicitly evaluate both no-tool and tool-augmented variants, allowing us to isolate the effect of tool use. Given $S_{\text{tr}}$, the model samples up to $k{=}5$ candidate Python programs under a fixed prompt (Fig. 5). We discard invalid or duplicate programs, execute the remaining candidates in a sandboxed interpreter, and select the program with the lowest validation error on $S_{\text{val}}$ (Fig. 2). We do not use validation feedback to adapt the sampling distribution.

**Baselines.** (1) *Classic ML:* XGBoost, Random Forest, SVM, and a Genetic Algorithm baseline, each with validation-based model and hyperparameter selection. (2) *A pre-trained tabular foundation model:* TabPFN (Hollmann et al., 2022). (3) *From scratch:* Qwen3-1.7B trained from scratch as a classifier on the $m{=}200$ labeled examples. (4) *Fine-tuning:* Qwen3-1.7B, Llama3.2-1B, and Deepseek-Coder-1.3B fine-tuned on the same $m{=}200$ labeled examples; hyperparameters (e.g., learning rate, batch size, and the number of tuned layers) are selected via a grid sweep. (5) *In-context learning:* We evaluate large language models prompted with all $m{=}200$ labeled examples and queried for the test labels. The comparison includes open-weight models, code-specialized models, and closed frontier models. Open-weight models are run without tool

| Task | n=20 | | n=25 | | n=30 | | n=50 | | n=100 | | PAC bound | |
|------|------|------|------|------|------|------|------|------|------|------|------|------|
| | Train | Test | Train | Test | Train | Test | Train | Test | Train | Test | ASCII | PyTok |
| **10-P** | 100 | 53.9 | 100 | 49.8 | 100 | 50.5 | 100 | 49.2 | 100 | 50.7 | 0.995 | 0.998 |
| **CAP** | 100 | 99.9 | 100 | 50.3 | 100 | 50.2 | 100 | 49.5 | 100 | 50.4 | 0.985 | 0.994 |
| **IsPrime** | 100 | 59.8 | 100 | 58.7 | 100 | 60.3 | 100 | 60.1 | 100 | 59.9 | 0.992 | 0.997 |

*Table 3.* **BLOOM-75M (SGD) train/test accuracy (%) and a program-length PAC bound for training with $m$=100k samples.** We report train/test accuracy at each length $n$. The last two columns report the bound $1 - m^{-1}[L \log |\Sigma| + \log(2L^2/\delta)]$ with $\delta$=$10^{-10}$, using $L$ from a compact reference implementation measured in ASCII bytes ($|\Sigma|$=128) or Python lexical tokens ($|\Sigma|$=64); the bound is independent of $n$. **Despite perfect training accuracy at each $n$, test accuracy stays near chance on 10-P and CAP, even though both admit short programs for which the PAC-learning bounds are close to 1.**

| Model | Adult Income | Secondary Mushroom | CDC Diabetes Health Indicators | HRTU2 | Chess (KR vs KP) |
|-------|-------------|-------------------|-------------------------------|-------|------------------|
| SVM | 52.8 ± 0.000 | 68.1 ± 0.000 | 69.1 ± 0.000 | 91.5 ± 0.000 | 89.4 ± 0.000 |
| GA | 75.2 ± 0.016 | 64.7 ± 0.020 | 69.0 ± 0.014 | 90.6 ± 0.001 | 92.5 ± 0.013 |
| Decision Tree | 67.8 ± 0.020 | 69.0 ± 0.016 | 62.5 ± 0.004 | 89.1 ± 0.002 | **94.1 ± 0.003** |
| Random Forest | 74.2 ± 0.008 | 70.8 ± 0.003 | **72.7 ± 0.006** | 91.4 ± 0.005 | 88.1 ± 0.011 |
| XGBoost | 71.4 ± 0.006 | 66.1 ± 0.012 | 68.6 ± 0.004 | 91.4 ± 0.001 | 93.1 ± 0.005 |
| TabPFN | **77.7 ± 0.000** | 69.9 ± 0.000 | 72.1 ± 0.000 | **93.1 ± 0.000** | 93.1 ± 0.000 |
| LLM-PV | 75.8 ± 0.007 | **72.2 ± 0.020** | 71.1 ± 0.002 | 92.3 ± 0.007 | 92.3 ± 0.012 |

*Table 4.* **Tabular benchmarks.** Test accuracy (%) for comparing classic ML baselines with LLM-PV. Our LLM-PV method achieves comparable results to other standard tabular learners.

use. For `GPT-5-Thinking`, we report both no-tool and tool-augmented variants; for `Gemini-3.1-Pro`, we report the tool-augmented variant. (6) *Agentic ML baselines:* We also compare against general-purpose agentic ML systems, including `MLAgentBench` (Huang et al., 2024) and `AIDE-ML` (Jiang et al., 2025). These baselines are instantiated with `GPT-5-Thinking` and tool use, allowing the agent to write, execute, and revise code during the search process. All prompting, tool-use, and evaluation details are provided in App. A.2.

### 5.3. Main results

**LLM baselines often fit without recovering the rule.** Across the core algorithmic tasks (parity variants, Dyck-2, CAP), both training from scratch and fine-tuning remain close to chance at $n$=100 (Tab. 1), suggesting length-specific memorization rather than rule recovery. Concretely, for the parity variants {FP, FHP, 3-P, 10-P}, SGD/FT scores fall in the range 48.6–54.3% at $n$=100 (Tab. 1); for CAP they are even tighter at 49.4–50.8% (Tab. 1). Fine-tuning can exploit local structure when it exists: on Pattern1 at $n$=100, the best FT model reaches 61.5% versus 51.5% for the best SGD model (Tab. 1). However, this improvement does not transfer to the parity variants or CAP, where FT remains near chance (Tab. 1). In-context learning at 30B scale yields occasional gains but still does not produce consistent algorithmic generalization: at $n$=100, ICL is the strongest baseline on SHA (up to 54.0%; Tab. 1), while remaining near chance on parity and CAP (Tab. 1).

**Tabular learners capture surface-level cues.** At $n$=100, tabular learners can succeed when local statistics are predictive (e.g., IsPal reaches 69.7–87.1% for SVM/GA; IsPrime reaches 51.7–61.8% for GA/XGBoost; Tab. 1). Notably, even TabPFN (Hollmann et al., 2022)—a strong pre-trained tabular foundation-model—does not close the gap on our algorithmic tasks: at $n$=100 it remains at (near) chance on FP/FHP/3-P/10-P, Dyck-2, and CAP (roughly 48–51% across these columns; Tab. 1). Overall, while tabular methods can exploit shallow cues when they exist, they fail on parity, Dyck-2, and CAP, highlighting a qualitative distinction between feature-based prediction and learning an algorithmic procedure.

**LLM-PV recovers algorithmic rules.** In contrast, LLM-PV achieves perfect accuracy at $n$=100 on all parity variants, IsPalindrome, Dyck-2, CAP, and IsPrime (all 100%; Tab. 1), indicating broad rule recovery rather than task-specific pattern fitting. More strikingly, LLM-PV maintains high accuracy on the shifted-primality task IsPrime+47, reaching 80.6% at $n$=100 (Tab. 1). Since adding a fixed offset breaks many superficial correlations while preserving a clean arithmetic relationship, this result suggests LLM-PV is not merely memorizing primality patterns but is often discovering a transferable computational structure. SHA remains difficult in our setup: all methods are close to chance, with the best performance coming from ICL (up to 54.0% at $n$=100; Tab. 1).

**Direct prompting is the wrong output space.** ICL asks the model to behave like the target function directly: given a new input, output a label. This is different from recovering an executable rule. Consistent with this distinction, even tool-augmented ICL remains unreliable: `GPT-5-Thinking` with tools improves over its no-tool variant on several tasks, such as FP (99.0% vs. 82.0%), FHP (88.0% vs. 66.0%), and 10-P (79.0% vs. 47.0%), but it still remains far from rule recovery on Pattern1, Pattern2, Dyck-2, CAP, CycleCount, and SHA (Tab. 1). Thus, tool-augmented direct prediction does not reliably produce algorithmic generalization.

**Tools help when they support program search.** The `GPT-5-Thinking` ablation inside LLM-PV shows that tool use is highly beneficial when it is embedded in a propose-and-verify loop. Without tools, LLM-PV recovers some structure but remains incomplete, e.g., 78.4% on Pattern1, 64.7% on Pattern2, and 79.9% on Dyck-2. With tools, the same backbone reaches 100% accuracy on 10 of the 13 tasks and remains high on IsPrime+47 (80.6%) and CycleCount (96.9%) (Tab. 1). This suggests that much of the gain comes from programmatic search: the agent can write code, execute it, inspect failures, test hypotheses, and refine candidate solutions.

**Tool use alone is not sufficient.** The comparison with ICL shows that tools are useful only when paired with the right search objective. Tool-augmented ICL has access to the same kind of external computation, but it still returns labels rather than an executable rule. In contrast, LLM-PV uses tools to search over programs and then selects among them using validation error. The gap between tool-augmented ICL and tool-augmented LLM-PV shows that the benefit is not merely tool access; it is tool use organized around program discovery.

**Generic agents are not enough; the output matters.** The agentic ML baselines `MLAgentBench` and `AIDE-ML`, both instantiated with `GPT-5-Thinking` and tool use, remain close to chance on most tasks. For example, `AIDE-ML` stays between roughly $49\%$ and $55.5\%$ across all tasks, while `MLAgentBench` also fails broadly and even drops below chance on several tasks (Tab. 1). These systems search over standard ML workflows and return trained predictors, whereas our tasks often require an explicit algorithmic rule. LLM-PV succeeds because its hypothesis class is executable programs, not generic predictors.

**Open-weight models show partial program discovery without tools.** The no-tool open-weight LLM-PV variants occasionally recover individual rules, but their successes are sparse and unstable across tasks. For example, `DeepSeek-Coder-33B-Instruct` and `OLMo-3-32B-Instruct` solve FP, `OLMo-3-32B-Instruct` reaches $83.5\%$ on Pattern2, and `Qwen3-30B-Instruct` reaches $80.4\%$ on Dyck-2 (Tab. 1). However, most other entries remain near chance, suggesting that the propose-and-verify formulation can expose partial rule-recovery ability without tools, but reliable recovery requires active testing and debugging.

**More data does not reliably fix SGD on these tasks.** To test whether these failures are a small-data artifact, we train `BLOOM-75M` from scratch with SGD on $m=100\text{k}$ examples per task (separately for each $n$). The model again fits the training data perfectly, yet test accuracy remains near chance on 10-P and on CAP for $n \geq 25$, and is only modest on IsPrime (Tab. 3). Fig. 4 further shows that scaling $m$ from 10 to 100k does not yield reliable generalization for the SGD baseline in our setup, whereas LLM-PV attains perfect test accuracy with $m=200$ on these tasks.

**Length generalization.** In Fig. 3 we study out-of-distribution length generalization by training on 200 samples at a short input length ($n=10$) and evaluating on longer sequences. We keep the SGD (and FT) setup identical to our primary experiments. The SGD and FT baselines are oftentime able to generalize at $n=10$ but fail to generalize across lengths: accuracy rapidly collapses toward $\approx 50\%$ as $n$ increases. In contrast, LLM-PV often synthesizes dimension-invariant programs, yielding stable performance across lengths, including perfect generalization on tasks such as Pattern Matching and Random 3-Parity.

**LLM-PV performs comparably to strong tabular ML baselines.** In addition to the tasks above, we also consider tabular datasets. For each task, we use only 100 fixed training samples and 100 fixed validation samples, and we report test error on the original test set. To ensure that the samples are not easily recognized by LLM-based methods (e.g., LLM-PV), we preprocess each input by applying a fixed random linear transformation $x \mapsto Wx + b$, where $W$ is diagonal and $b$ is a vector, with entries drawn i.i.d. from a standard normal distribution. As per Tab. 4, LLM-PV is competitive against standard tabular learners, suggesting that propose-and-verify can remain effective beyond synthetic sequence tasks.

## 6. Discussion

We studied LLM-PV, a learning algorithm that uses a pretrained LLM as a proposal prior and selects among executable programs by validation error. Across algorithmic tasks, LLM-PV often recovers exact rules from few examples and generalizes far beyond the training length. This suggests a different use of pretrained LLMs in learning: not as predictors that directly imitate the target function, but as search priors over discrete, verifiable hypotheses.

The main lesson is that the output space matters. Standard gradient-based predictors, fine-tuned models, and in-context learning ultimately produce labels or learned predictors. By contrast, LLM-PV searches over executable rules and uses held-out error only to select among candidate programs. This separation keeps the learning criterion simple and auditable: the LLM proposes programs, while execution and validation performance decide.

More broadly, our results point toward *learning via LLM-guided search with verification*. Future work should characterize when the LLM proposal distribution places enough mass on correct or near-correct programs, when validation can reliably identify the right hypothesis, and where the approach breaks down.

The main limitation is scope. Our experiments focus on structured low-data regimes, and LLM-PV is not intended as a general replacement for supervised deep learning. Its success depends on a capable agentic proposer and on sufficient proposal mass near the correct program. As target programs become longer, more compositional, or less aligned with the pretrained model's prior, this mass may decrease, requiring more samples, more LLM calls, or richer propose-verify-refine procedures. Extending the approach to larger synthesis problems and real-world structured domains remains future work.

## Impact Statement

This work studies program learning in controlled algorithmic settings. It raises no direct ethical, safety, or environmental concerns in the scope considered here. We introduce LLM-PV, a propose-and-verify framework that performs ERM-style selection over a discrete program class using a pretrained model only as a proposal prior, and we provide empirical evidence that it can recover compact rules from few labeled examples and generalize to longer inputs. These results inform how pretrained priors can support auditable, executable hypotheses without gradient-based retraining. While our experiments execute candidate programs, they do so in a restricted setting; any application beyond this scope should use sandboxed execution and a constrained language to avoid unintended behavior.

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

# A. Additional Experimental Details And Results

## A.1. Evaluation Tasks

We evaluate across a suite of synthetic algorithmic tasks that isolate distinct forms of structure: sparse global rules (parity), position-invariant search (pattern matching), symmetry (palindromes), context-free constraints (Dyck-2), arithmetic (primality), and compositions of local nonlinear maps with global summaries (CAP, SHA). Unless noted otherwise, for each task and input length $n$ we construct class-balanced datasets with equal numbers of positive and negative examples.

- **Parity (FP/FHP/$k$-P).** Inputs are $x \in \{0, 1\}^n$ and labels are $\pm 1$. For a fixed selector $s \in \{0, 1\}^n$, define $y(x) = (-1)^{\langle s, x \rangle}$. We consider: (i) *full parity* (FP) with $s = \mathbf{1}_n$; (ii) *first-half parity* (FHP) with $s = (\mathbf{1}_{n/2} \| \mathbf{0}_{n/2})$ (for even $n$); and (iii) *random $k$-parity* ($k$-P), where $s$ is sampled uniformly from $\{0, 1\}^n$ subject to $\|s\|_0 = k$ (fixed per run). Datasets are generated by sampling $x \sim \mathrm{Unif}(\{0, 1\}^n)$ and labeling by $y(x)$.

- **Pattern Matching (Pattern1/Pattern2).** Fix a binary pattern $p \in \{0, 1\}^k$ with $k < n$ and input $x \in \{0, 1\}^n$. The label is $y(x) = \mathbb{I}\big[\exists i \in \{1, \dots, n - k + 1\} \text{ s.t. } (x_i, \dots, x_{i+k-1}) = p\big]$. We use $p = 00111111$ (Pattern1) and $p = 10101010$ (Pattern2). This task requires a position-invariant "search" rule rather than reliance on fixed coordinates.

- **Palindrome (IsPal).** For $x \in \{0, 1\}^n$, $y(x) = \mathbb{I}\big[\forall i \in \{1, \dots, \lfloor n/2 \rfloor\} : x_i = x_{n-i+1}\big]$. Positives are constructed by sampling a random first half and mirroring it. Negatives are generated by first constructing a palindrome and then flipping one bit in the first half; this produces "near-miss" negatives that share most local statistics with positives.

- **Dyck-2.** Let $\mathcal{M} : \{0, 1\}^2 \to \{\,(,\,),\,[,\,]\,\}$ map bit-pairs to bracket symbols, and let $S(x)$ be the resulting bracket string for $x \in \{0, 1\}^n$ (so $|S(x)| = n/2$ when $n$ is even). The label is $y(x) = \mathbb{I}[S(x) \in D_2]$, where $D_2$ is the Dyck-2 language of balanced parentheses with two bracket types. This probes recognition of a context-free constraint that naturally admits stack-like reasoning.

- **Primality with restricted last digit (IsPrime).** Inputs are digit strings $x = (x_1, \dots, x_n) \in \{0, \dots, 9\}^n$ interpreted as an $n$-digit base-10 integer. The label is $y(x) = \mathbb{I}\big[\mathrm{IsPrime}(\mathrm{int}(x))\big]$. We construct balanced datasets by sampling uniformly from the set of $n$-digit primes (positives) and the set of $n$-digit composite numbers (negatives), while excluding leading zeros.[1] We constrain the data distribution to integers whose last digit lies in $\{1, 3, 7, 9\}$, i.e., $x_n \in \{1, 3, 7, 9\}$, removing the most common last-digit shortcut and forcing dependence on the full input.

- **Cellular Automaton Parity (CAP).** Inputs are $x \in \{0, 1\}^n$. Define a one-step local update $x \mapsto x'$ with boundary conditions $x_0 = x_{n+1} = 0$: $x_i' = x_{i-1} \oplus (x_i \vee x_{i+1})$ ($i = 1, \dots, n$). The label is the parity of the updated string, $y(x) = (\sum_{i=1}^n x_i') \bmod 2$. This composes a local nonlinear transformation with a global summary statistic.

- **Graph Cycle Density (CountCycles, fn_aa).** Inputs are graphs encoded as strings of edges like `u12v23u45v67...`, where each edge is `u` + two-digit source + `v` + two-digit destination. All instances satisfy $V = E = n/2$, so the cycle rank $E - V + C(G)$ equals the number of connected components $C(G)$. We label by $y(G) = \mathbb{I}[C(G) > E/6]$. Positives have many small components (e.g., triangles and disjoint edges), while negatives have one or a few large components (spanning tree plus extra edges). Node IDs are permuted and edges are shuffled to avoid positional shortcuts. Whenever necessary (running `TabPFN`), we tokenize the characters as follows: digits 0–9 $\mapsto$ 0–9, `u` $\mapsto$ 10, `v` $\mapsto$ 11.

- **SHA-256 Parity (SHA).** For $x \in \{0, 1\}^n$, compute the 256-bit digest $(h_1, \dots, h_{256}) = \mathrm{SHA}\text{-}256(x)$ and output its parity: $y(x) = (\sum_{i=1}^{256} h_i) \bmod 2$. Because cryptographic hashes behave pseudorandomly, this is intended as a stringent stress test.

## A.2. Baselines

We compare LLM-PV to four families of baselines: (i) classic tabular classifiers, (ii) LMs trained from scratch with SGD, (iii) fine-tuning pre-trained LMs, and (iv) in-context prediction with large instruction-tuned LMs. Unless stated otherwise, experiments use $m = 200$ labeled examples, split into $|S_{\mathrm{tr}}| = 100$ and $|S_{\mathrm{val}}| = 100$, and are evaluated on an independent test set of 10,000 examples.

---

[0]For Dyck-2 at $n = 20$, the number of valid strings is too small to sample the standard-sized balanced test set without excessive duplication; in this case we reduce the test set to 1,000 examples.

[1]We exclude leading zeros when sampling $n$-digit integers.

**Problem Statement:** Given a sequence of input vectors (binary, length {sequence_dimension}) and their corresponding scalar binary outputs ('0' or '1'), find a concise Python function `f(x)` that accurately approximates the underlying relationship. The function should not be a trainable model, but a direct logical or mathematical representation of the target function.

**Data Examples:**

```
000111101011110010100101001100 -> 1
... 011011010111000010010101001000 -> 1
```

**You must output ONLY a single JSON object:**

```
{"code": "<python function>"}
```

*Figure 5.* Prompt used in our LLM-PV procedure. We run `GPT-5` with this prompt for up to $k$ independent iterations, each returning only Python code for a candidate target function.

**Classic ML baselines.** We train XGBoost, Random Forest, SVM, and a Genetic Algorithm (GA) baseline on $S_{\text{tr}}$ and select hyperparameters by validation performance on $S_{\text{val}}$; reported numbers use the selected configuration and are evaluated on the test set. Hyperparameter grids are: (i) **SVM**: $C \in \{0.1, 1, 10, 100\}$, $\gamma \in \{\texttt{scale}, \texttt{auto}, 0.001, 0.01, 0.1\}$, $\texttt{kernel} \in \{\texttt{rbf}, \texttt{poly}, \texttt{sigmoid}\}$; (ii) **Random Forest**: $\texttt{n\_estimators} \in \{64, 128, 256\}$, $\texttt{max\_depth} \in \{5, 10, 15, \texttt{None}\}$, $\texttt{min\_samples\_split} \in \{2, 5, 10\}$; (iii) **XGBoost**: $\texttt{n\_estimators} \in \{100, 128, 256\}$, $\texttt{max\_depth} \in \{5, 6, 7\}$, $\texttt{learning\_rate} \in \{0.1, 0.3\}$, $\texttt{subsample} \in \{0.8, 1.0\}$; (iv) **Genetic Algorithm (GA)**: genetic programming over Boolean expression trees with leaves given by input variables or Boolean constants and internal nodes drawn from $\{\texttt{NOT}, \texttt{XOR}, \texttt{AND}, \texttt{OR}\}$; prediction is obtained by evaluating the tree on the input bits. Fitness is training accuracy (optionally on a random subsample for speed) minus a small size penalty to discourage large expressions. We evolve the population using tournament selection, subtree crossover, subtree mutation, and elitism, and return the best individual (selected by training fitness; reported with validation/test accuracy). GA hyperparameters are: population size $\texttt{pop\_size} \in \{300, 500\}$, number of generations $\texttt{generations}=80$, initial program depth $\texttt{max\_depth\_init} \in \{6, 10\}$, mutation depth cap $\texttt{max\_depth\_mut} \in \{4, 6\}$, crossover probability $\texttt{cx\_prob}=0.7$, and mutation probability $\texttt{mut\_prob}=0.25$.

**Training from scratch (SGD).** We train decoder-only language-model backbones from scratch as binary classifiers on sequence-encoded pairs $(x, y)$, minimizing binary cross-entropy with AdamW (Loshchilov & Hutter, 2019). Our SGD baselines are `Qwen3-1.7B` (Yang et al., 2025), `Llama3.2-1B` (Meta AI, 2024), and `Deepseek-Coder-1.3B` (Guo et al., 2024). We adapt each LM to classification by (i) restricting the tokenizer vocabulary to three tokens (`vocab_size=3`) and (ii) replacing the LM head with a single-logit linear classifier (hidden$\rightarrow$1). Unless stated otherwise, we train for 200 epochs with batch size 20 using cosine-annealed learning rates with $\eta_{\max}=10^{-5}$ and $\eta_{\min}=10^{-6}$. To probe data scaling with a lighter model, we additionally train `BLOOM-75M`, a scaled-down BLOOM-style architecture (Scao et al., 2023) (hidden size 512, 8 heads, 24 layers), on $m=100\text{k}$ examples per (task, $n$) for 1,000 epochs with batch size 256 and constant learning rate $\eta=10^{-5}$.

**Fine-tuning pre-trained LMs (FT).** We fine-tune `Qwen3-1.7B` (Yang et al., 2025), `Llama3.2-1B` (Meta AI, 2024), and `Deepseek-Coder-1.3B` (Guo et al., 2024). For each model and input length $n \in \{20, 25, 30, 50, 100\}$, we train with AdamW for 1,000 epochs using a cosine-annealing learning-rate schedule, and report accuracy on 10k held-out test examples. We evaluate both full fine-tuning and partial fine-tuning of only the top $\{2, 4, 8\}$ transformer blocks. Hyperparameters are selected via grid search over the number of tuned layers, batch size $\{20, 50, 100\}$, and learning rate $\{5 \times 10^{-3}, 10^{-3}, 5 \times 10^{-4}, 10^{-4}, 5 \times 10^{-5}, 10^{-5}\}$, choosing the configuration with the best test performance. The selected batch size is 20 for all models, with peak learning rates of $10^{-3}$ for `Llama3.2-1B` and $5 \times 10^{-3}$ for `Qwen3-1.7B` and `Deepseek-Coder-1.3B`; the scheduler anneals to a minimum learning rate of $10^{-3}$. All runs use `bfloat16`, space-separated integer tokenization with EOS padding, and a single-logit classifier head applied to the final position.

**In-context learning (ICL).** We evaluate whether a large instruction-tuned LLM can recover the latent rule from labeled examples in its context window and apply it to a new input. Concretely, given a training set $S_{\text{tr}}$ of 200 labeled pairs, we ask whether the induced predictor $h(x_{\text{test}} \mid S_{\text{tr}})$ matches the ground-truth label $y(x_{\text{test}})$. We use `Qwen3-30B-A3B-Instruct-2507` and `Qwen3-Coder-30B-A3B-Instruct` (Yang et al., 2025), and `Deepseek-Coder-33B-Instruct` (Guo et al., 2024). For each of 100 test inputs, we construct a prompt (Fig. 6) containing the task description, the full set of 200 training examples, and the held-out test input, and then query the model for the corresponding label. We decode with temperature 0.2, top-$p$ 0.95, and a maximum of 1024 generated tokens.

---

**LLM Prompt**

**Problem Statement:** Given a sequence of input vectors (binary, length {sequence_dimension}) and their corresponding scalar binary outputs ('0' or '1'), you have to learn a hypothesis that approximates the underlying relationship. Given the data below, determine what is the label for the given string and output ONLY the label. **Data Examples:**

```
0001111010111100101001010011 00 -> 1
... 011011010111000010010101001000 -> 1
```

**Test Input:**

```
0101001101110010010101101001000
```

**You must output ONLY a single JSON object: {"lable": "¡your predicted label¿"}**

---

*Figure 6.* Prompt used in in-context learning procedure. We run three models `Qwen3-30B-A3B-Instruct-2507`, `Qwen3-Coder-30B-A3B-Instruct`, and `Deepseek-Coder-33B-Instruct` with this prompt. For each prompt, the model outputs only the predicted label for the test input.

| Task | Model | Accuracy vs. training set size | | | | | | | | | | | | | | | | |
|---|---|---|---|---|---|---|---|---|---|---|---|---|---|---|---|---|---|---|
| | | m=10 | | | m=100 | | | m=200 | | | m=1,000 | | | m=10,000 | | | m=100,000 | | |
| | | Train | Test | Bound (A/T) | Train | Test | Bound (A/T) | Train | Test | Bound (A/T) | Train | Test | Bound (A/T) | Train | Test | Bound (A/T) | Train | Test | Bound (A/T) |
| **10-P** | BLOOM-75M | 100% | 49.9% | 0.0/0.0% | 100% | 49.1% | 0.0/0.0% | 100% | 49.7% | 0.0/0.0% | 100% | 49.9% | 46.7/76.5% | 99.9% | 50.1% | 94.7/97.6% | 99.8% | 50.7% | 99.5/99.8% |
| | LLM-PV | 100% | 49.3% | 0.0/0.0% | 100% | 100% | 0.0/0.0% | 100% | 100% | 0.0/0.0% | – | – | 46.7/76.5% | – | – | 94.7/97.6% | – | – | 99.5/99.8% |
| **IsPrime2** | BLOOM-75M | 100% | 50.9% | 0.0/0.0% | 100% | 50.2% | 0.0/0.0% | 100% | 51.3% | 0.0/0.0% | 100% | 56.4% | 22.9/71.9% | 100% | 59.7% | 92.3/97.2% | 100% | 59.9% | 99.2/99.7% |
| | LLM-PV | 100% | 50.5% | 0.0/0.0% | 100% | 100% | 0.0/0.0% | 100% | 100% | 0.0/0.0% | – | – | 22.9/71.9% | – | – | 92.3/97.2% | – | – | 99.2/99.7% |
| **CAP** | BLOOM-75M | 100% | 50.1% | 0.0/0.0% | 100% | 50.4% | 0.0/0.0% | 100% | 50.4% | 0.0/0.0% | 100% | 50.0% | 0.0/42.2% | 99.6% | 50.0% | 85.5/94.2% | 99.9% | 50.4% | 98.5/99.4% |
| | LLM-PV | 100% | 50.3% | 0.0/0.0% | 100% | 50.1% | 0.0/0.0% | 100% | 100% | 0.0/0.0% | – | – | 0.0/42.2% | – | – | 85.5/94.2% | – | – | 98.5/99.4% |

*Table 5.* **Scaling labeled data does not reliably fix SGD generalization, even on tasks with short programs.** We report train/test accuracy (%) at $n=100$ as a function of the number of labeled training examples $m$ (test set size is 10k in all cases). `BLOOM-75M` trained with SGD attains (near-)perfect training accuracy throughout, yet test accuracy remains near chance on **10-P** and **CAP**, and improves only modestly on **IsPrime2**. LLM-PV is far more sample-efficient in this setup, solving **10-P** and **IsPrime2** with $m \leq 100$ and all three tasks with $m=200$. The **Bound (A/T)** columns give the length-based lower bound $1 - m^{-1}[L \log |\Sigma| + \log(2L^2/\delta)]$ (with $\delta = 10^{-10}$), evaluated using compact reference implementations with length $L$ measured in **ASCII bytes** (A, $|\Sigma|=128$) or **Python lexical tokens** (T, $|\Sigma|=64$ token types); values are clipped to $[0, 1]$ and reported as **A/T**. This bound depends only on $(m, L, |\Sigma|, \delta)$ and is included to emphasize that these targets admit short descriptions, even though SGD does not exploit this bias here.

**LLM-PV.** LLM-PV (Alg. 1, Fig. 2) uses a pretrained code LLM (`GPT-5-Thinking`) as a proposal distribution over candidate programs. We split $S$ into $S_{\text{tr}}$ and $S_{\text{val}}$ of equal size, build a prompt from $S_{\text{tr}}$ (Fig. 5), sample up to $k=5$ candidate programs, compile and filter invalid/duplicate outputs, and select the candidate with minimum validation error on $S_{\text{val}}$. We run the API with `reasoning_effort=High`, `text_verbosity=Low`, `max_tokens=20k`, and a per-call timeout of 20 minutes; sampling parameters such as temperature/top-$p$ are platform-managed.

### A.3. Additional Results

#### A.3.1. DATA SCALING

To investigate the impact of training set size on generalization, we trained the `BLOOM-75M` baseline using SGD across multiple data scales, using $m_{\text{train}} \in \{10, 100, 200, 1000, 10k, 100k\}$ for $n=100$. This evaluation was conducted on Random 10-Parity, Cellular Automata Parity, and IsPrime (with restricted negatives). The `BLOOM-75M` model was trained for 1000 epochs and a constant learning rate of $\eta = 10^{-5}$. The batch size was set to 256 for training sets with 1000 or more samples, and to 10 for smaller datasets. The model trained on this largest dataset was also evaluated across multiple input lengths $n \in \{20, 25, 30, 50, 100\}$.

The results highlight a failure of the SGD baseline to generalize, even with extensive data. As detailed in Fig. 4 and Tab. 5, increasing the training set size did not yield better test performance; the model consistently overfits, achieving perfect training accuracy while its test accuracy on Random 10-Parity and Cellular Automata Parity remained at the $\approx 50\%$ chance level. Furthermore, Tab. 3 show that even with 100k training samples, the model still overfits when the input dimension is too high, with performance again collapsing to chance. In contrast, LLM-PV seems to be sample-efficient, synthesizing correct programs for all three tasks with a training set size of just 100–200.

> **Prompt Templates for Prompt Ablation**
>
> **P1.** Given a sequence of input vectors ({data_mode}, length {seq_len}) mapped to scalar binary outputs, extract the underlying relationship as a concise Python function `f(x)`. The solution must be a direct logical or mathematical expression, not a machine learning model.
>
> **P2.** Analyze the provided input vectors ({data_mode}, length {seq_len}) and their corresponding binary outputs to determine the governing logic. Express this logic as a short, deterministic Python function `f(x)` using mathematical or logical operations, avoiding trainable parameters.
>
> **P3.** Identify the mapping between the input vectors ({data_mode}, length {seq_len}) and binary scalar outputs. Represent this mapping through a concise, stateless Python function `f(x)` that relies only on explicit mathematical or logical rules rather than learned weights.
>
> **P4.** Discover the strict mathematical or logical rule that maps the input vectors ({data_mode}, length {seq_len}) to their binary outputs. Output a concise Python function `f(x)` that formally encodes this rule without relying on any trainable architecture.
>
> **P5.** Infer the rule that best explains the observed input-output pairs, where each input is a vector in {data_mode} of length {seq_len} and each output is binary. Return the rule as a compact Python function `f(x)` using explicit logical or mathematical operations.

*Table 6.* **Prompt templates used in the prompt ablation experiments.** Here, {data_mode} denotes the input representation and {seq_len} denotes the sequence length, with $n = 100$.

### A.3.2. HYPERPARAMETER ABLATION

To assess whether SGD's failures are driven by particular optimization settings, we ran learning-rate and batch-size sweeps in the 200-sample regime, holding all other choices fixed. We trained `Qwen3-1.7B` for 200 epochs on 200 training samples and evaluated on 10k random test samples, across three representative tasks (10-P, Pattern2, IsPal) and lengths up to $n=100$. For the learning-rate sweep, we used batch size $B=20$ and varied $\eta \in \{8 \times 10^0, 8 \times 10^{-1}, \ldots, 8 \times 10^{-7}\}$. For the batch-size sweep, we fixed $\eta$ and varied $B \in \{10, 20, 50, 100, 200\}$. Tab. 9 reports, for each (task, $n$), the best test accuracy achieved over the corresponding sweep.

Across both sweeps, the qualitative picture is unchanged. Random 10-Parity (10-P) stays at chance for all $n$, even after selecting the best $\eta$ or the best $B$ (max $\leq 51.1\%$), indicating that no setting in these ranges yields meaningful generalization. Pattern Matching (Pattern2) can reach near-perfect accuracy at short lengths (e.g., 100% at $n=20$ when sweeping $\eta$), but performance drops sharply with increasing $n$ and falls to 56.9% at $n=100$ even after optimizing over $\eta$ (and to 65.5% after optimizing over $B$). Similarly, IsPalindrome (IsPal) improves at short lengths under the best settings (up to 72.4% at $n=20$), yet collapses to chance by $n=100$ (max 50.0% over $\eta$ and 49.6% over $B$).

Overall, tuning learning rate or batch size mainly affects short-length performance and does not prevent the systematic degradation with length. This supports the interpretation that the dominant failure mode is out-of-distribution length generalization, rather than a fixable optimization detail.

### A.3.3. PROMPT ABLATION

We evaluate the sensitivity of LLM-PV to the exact wording of the prompt given to `GPT-5-Thinking`. To construct the ablation, we use five prompt templates that express the same high-level instruction: infer the relationship from the training samples and return a concise Python function `f(x)` that implements the rule. The prompts differ only in phrasing, not in the information provided to the model.

Tab. 7 reports the performance of LLM-PV under each prompt. The results are stable across prompt variations. In particular, all prompts achieve perfect performance on PATTERN2, 10-P, ISPRIME, and CAP. On the hardest task, ISPRIME+47, performance varies mildly across prompts, with an average accuracy of $82.5 \pm 3.1$. These results suggest that the gains of LLM-PV are not driven by a carefully engineered prompt.

| Prompt | Pattern2 | 10-P | isPrime | CAP | isPrime+47 |
|--------|----------|------|---------|-----|------------|
| **P1** | 100 | 100 | 100 | 100 | 81.8 |
| **P2** | 100 | 100 | 100 | 100 | 77.9 |
| **P3** | 100 | 100 | 100 | 100 | 86.7 |
| **P4** | 100 | 100 | 100 | 100 | 83.0 |
| **P5** | 100 | 100 | 100 | 100 | 83.0 |
| **Average** | $100 \pm 0$ | $100 \pm 0$ | $100 \pm 0$ | $100 \pm 0$ | $82.5 \pm 3.1$ |

*Table 7.* **Prompt variation results for LLM-PV.** We report test accuracy across five prompt templates. Performance is stable across prompt rephrasings, indicating that the method is not sensitive to a single carefully engineered prompt.

| Method | Train time (seconds per sample) | Test time (seconds per sample) |
|--------|----------------------------------|---------------------------------|
| **SGD: Qwen3-1.7B** | 0.671 | 0.016 |
| **FT: Qwen3-1.7B** | 0.345 | 0.017 |
| **LLM-PV: GPT-5** | 2.091 | 0.224 |
| **ICL: GPT-5-Thinking** | 0.001 | 53.961 |
| **ICL: Qwen3-30B-Instruct** | 0.001 | 0.139 |

*Table 8.* **Wall-clock runtime.** Train and test times are reported in seconds per sample. LLM-PV has higher training cost due to LLM-based proposal generation, but its selected program can be evaluated efficiently at test time.

### A.3.4. WALL-CLOCK TIMES

We report the wall-clock runtime of the main baselines and LLM-PV in Tab. 8. The comparison includes SGD training from scratch and fine-tuning on `Qwen3-1.7B`, LLM-PV with `GPT-5`, and in-context learning with `GPT-5-Thinking` and `Qwen3-30B-Instruct`. All times are normalized per sample, computed as the total runtime divided by the number of samples. For SGD training from scratch and fine-tuning, we report the runtime using the minimum number of epochs required to reach $100\%$ training accuracy.

The results highlight the main computational tradeoff. LLM-PV incurs the largest training-time cost, primarily because program proposals are generated by an LLM. However, once a program has been selected, inference is fast: its test-time cost is substantially lower than `GPT-5-Thinking` in-context learning and competitive with the other non-API baselines. Gradient descent and fine-tuning are fast at test time, but their predictive performance is poor. In-context learning avoids training altogether, but its test-time cost can be large because every test example requires a fresh model call. Thus, LLM-PV shifts computation from test time to training time and obtains the best performance while retaining efficient deployment.

### A.4. LLM Reasoning Traces

We extend (Fig. 7) the reasoning-trace analysis from Fig. 1 to Cellular Automata Parity and Full Parity to highlight how the model adapts its search strategy across tasks.

For Cellular Automata Parity, the model starts with simple dataset checks and parity-style baselines, then attempts a linear rule over $\mathbb{F}_2$. When this fails, it escalates to a structured search over non-linear, hand-designed features (for example edge bits, transition counts such as #01/#10, and their parities), and identifies an XOR rule that fits perfectly. It then rewrites the rule using parity identities and verifies equivalent forms.

For Full Parity, the trace is much shorter: after a quick sanity check, the model proposes global parity and immediately verifies a perfect match. This contrast shows that the model expands its hypothesis class only when simpler families fail, but locks onto the correct rule quickly when it is obvious.

| Task | Summary | $n{=}20$ | $n{=}25$ | $n{=}30$ | $n{=}50$ | $n{=}100$ |
|------|---------|--------|--------|--------|--------|---------|
| **10-P** | max over $\eta$ | 50.4 | 50.3 | 50.4 | 50.2 | 51.1 |
| | max over $B$ | 50.8 | 50.4 | 50.4 | 50.9 | 50.4 |
| **Pattern2** | max over $\eta$ | 100.0 | 94.8 | 93.9 | 87.9 | 56.9 |
| | max over $B$ | 97.7 | 96.4 | 96.0 | 89.5 | 65.5 |
| **IsPal** | max over $\eta$ | 72.4 | 60.5 | 54.5 | 50.3 | 50.0 |
| | max over $B$ | 68.6 | 57.5 | 53.9 | 50.8 | 49.6 |

*Table 9.* **Hyperparameter ablations (test accuracy, %).** For each task and sequence length $n$, we report the maximum test accuracy obtained by sweeping learning rates $\eta \in \{8 \times 10^0, \ldots, 8 \times 10^{-7}\}$ and batch sweep over $B \in \{10, 20, 50, 100, 200\}$. 10-P remains near chance across all $n$, while Pattern2 and IsPal achieve high accuracy at short lengths but degrade substantially as $n$ increases.

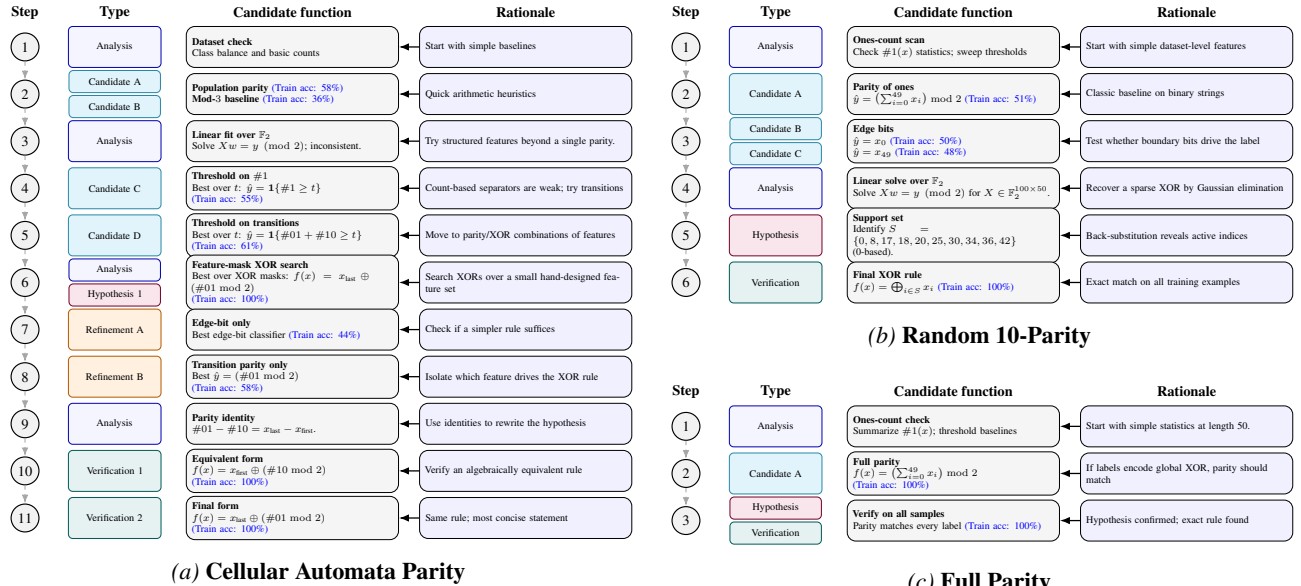

*Figure 7.* **Reasoning traces for discrete synthetic tasks.** Left: Cellular Automata Parity, where the search escalates from simple heuristics to an XOR rule over hand-designed features and then simplifies it via parity identities. Right (top): Random 10-Parity, solved by a linear system over $\mathbb{F}_2$ that recovers the active XOR support. Right (bottom): Full Parity, identified directly by the global parity hypothesis and verified on all samples.

## B. Proofs

### B.1. Proof of Eq. 1

**Theorem B.1** (Valiant (1984); see also Cor. 2.3 of Shalev-Shwartz & Ben-David (2014)). *Let $y : \mathcal{X} \to \{\pm 1\}$ be an unknown target function and let $\mathcal{H} \subset \{\pm 1\}^{\mathcal{X}}$ be a finite hypothesis class. Suppose we are in the realizable setting (i.e., $y \in \mathcal{H}$). Let $S = \{(x_i, y(x_i))\}_{i=1}^m$ be $m$ training examples drawn i.i.d. from a distribution $D$ over $\mathcal{X} \times \{\pm 1\}$. Then, with probability at least $1 - \delta$ over the draw of $S$, every hypothesis $h \in \mathcal{H}$ that is consistent with $S$ satisfies*

$$\mathrm{err}_D(h) \;\leq\; \frac{\log(|\mathcal{H}|) + \log(1/\delta)}{m}.$$

**Corollary B.2.** *Let $y : \mathcal{X} \to \{\pm 1\}$ be an unknown target function and let $\mathcal{H} = \bigcup_{\ell \geq 1} \mathcal{H}_\ell \subset \{\pm 1\}^{\mathcal{X}}$ be a union of finite sets. Suppose we are in the realizable setting (i.e., $y \in \mathcal{H}$). Let $S = \{(x_i, y(x_i))\}_{i=1}^m$ be $m$ training examples drawn i.i.d. from a distribution $D$ over $\mathcal{X} \times \{\pm 1\}$. Then, with probability at least $1 - \delta$ over the draw of $S$, for any $\ell \in \mathbb{N}$ and any hypothesis $h \in \mathcal{H}_\ell$ that is consistent with $S$, it holds that*

$$\mathrm{err}_D(h) \;\leq\; \frac{\log(|\mathcal{H}_\ell|) + \log\big((\pi^2/6)\,\ell^2/\delta\big)}{m}.$$

---

**Algorithm 2** Length-First Program Search (LFPS)

---

**Require:** Sample $S = \{(x_i, y_i)\}_{i=1}^m$, language $\mathcal{L} \subseteq \Sigma^*$, per-run timeout $T \in \mathbb{N}$, optional max length $L_{\max} \in \mathbb{N} \cup \{\infty\}$

**Ensure:** A program $u^\star \in \mathcal{L}$ whose total semantics $[\![u^\star]\!] : \mathcal{X} \to \{\pm 1\}$ satisfies $[\![u^\star]\!](x_i) = y_i$ for all $(x_i, y_i) \in S$; or $\perp$ if none is found up to $L_{\max}$

1: **for** $\ell = 1, 2, \ldots, L_{\max}$ **do**
2:    **for all** strings $u \in \mathcal{L}$ with $|u| = \ell$ in lexicographic order **do**
3:       **if** $u$ fails to compile **then continue**
4:       $consistent \leftarrow$ true
5:       **for** each $(x_i, y_i) \in S$ **do**
6:          Run $u$ on input $x_i$ for at most $T$ steps; let $o_i \in \{\pm 1, \perp\}$ be the output ($\perp$ if no halt)
7:          **if** $o_i = \perp$ **or** $o_i \neq y_i$ **then**
8:             $consistent \leftarrow$ false; **break**
9:          **end if**
10:      **end for**
11:      **if** $consistent$ **then**
12:         **return** $u^\star \leftarrow u$                 {minimal-length consistent program}
13:      **end if**
14:    **end for**
15: **end for**
16: **return** $\perp$                         {no consistent total program found up to $L_{\max}$}

---

*Proof.* Assume that $y \in \mathcal{H} = \bigcup_{\ell \geq 1} \mathcal{H}_\ell$; hence, there exists some $\ell^*$ for which $\mathcal{H}_{\ell^*}$ is realizable. For each fixed $\ell$ with at least one hypothesis consistent with $S$, Thm. B.1 implies that for any $\delta_\ell > 0$, with probability at least $1 - \delta_\ell$, every $h \in \mathcal{H}_\ell$ consistent with $S$ satisfies

$$\mathrm{err}_D(h) \leq \frac{\log(|\mathcal{H}_\ell|) + \log(1/\delta_\ell)}{m}.$$

Choose $\delta_\ell = \frac{6}{\pi^2} \frac{\delta}{\ell^2}$, so that $\sum_{\ell \geq 1} \delta_\ell = \delta$. Then, for each such $\ell$, with probability at least $1 - \delta_\ell$,

$$\mathrm{err}_D(h) \leq \frac{\log(|\mathcal{H}_\ell|) + \log\big((\pi^2/6)\, \ell^2/\delta\big)}{m}.$$

Applying a union bound over all $\ell \geq 1$ yields the claim (for each $\ell$ with no consistent hypothesis, the inequality is vacuous). $\qquad\square$

**Proposition B.3.** *Suppose we wish to learn a target function $y : \mathcal{X} \to \{\pm 1\}$ that can be implemented as a program of length $L$ in a programming language $\mathcal{L}$. Let $\mathcal{L}_\ell$ denote the set of programs of length $\ell$ in $\mathcal{L}$, and let $S = \{(x_i, y(x_i))\}_{i=1}^m$ be $m$ training examples drawn i.i.d. from a distribution $D$ over $\mathcal{X} \times \{\pm 1\}$. Then, with probability at least $1 - \delta$ over the draw of $S$, Alg. 2 outputs a program $h \in \mathcal{L}$ that is consistent with $S$ and satisfies*

$$\mathrm{err}_D(h) \leq \frac{L \log |\Sigma| + \log(2L^2/\delta)}{m}.$$

*Proof.* Since $y \in \mathcal{L}$, there exists a minimal length $L$ such that $y \in \mathcal{L}_L$. Therefore, there is at least one program of length $L$ consistent with $S$. Alg. 2 enumerates programs in order of increasing length, so it eventually returns a program $h$ of some length $\ell \leq L$ that is consistent with $S$. Every program in $\mathcal{L}_\ell$ is described over the alphabet $\Sigma$, hence $|\mathcal{L}_\ell| \leq |\Sigma|^\ell$ and $\log |\mathcal{L}_\ell| \leq \ell \log |\Sigma| \leq L \log |\Sigma|$. Applying Cor. B.2 with $\mathcal{H} = \mathcal{L}$ and $\mathcal{H}_\ell = \mathcal{L}_\ell$ and then upper-bounding by $L$ gives

$$\mathrm{err}_D(h) \leq \frac{\log |\mathcal{L}_\ell| + \log(2\ell^2/\delta)}{m} \leq \frac{L \log |\Sigma| + \log(2L^2/\delta)}{m}.$$

$\qquad\square$

**Proposition 4.1.** *Fix $\epsilon \geq 0$ and $\delta \in (0, 1)$. Draw independent samples $S_{\mathrm{tr}} \sim D^{m_{\mathrm{tr}}}$ and $S_{\mathrm{val}} \sim D^{m_{\mathrm{val}}}$ with labels from the target $y : \mathcal{X} \to \{\pm 1\}$. Run Alg. 1 for $k$ trials, and assume each trial outputs an accepted total hypothesis $h_t : \mathcal{X} \to \{\pm 1\}$*

*using only $S_{\mathrm{tr}}$. Let $h^\star \in \arg\min_{t \in [k]} \mathrm{err}_{S_{\mathrm{val}}}(h_t)$ be the returned hypothesis. If $k \geq \lceil \frac{\log(\delta/2)}{\log\left(1 - p_\epsilon(S_{\mathrm{tr}})\right)} \rceil$, then with probability at least $1 - \delta$ over $(S_{\mathrm{tr}}, S_{\mathrm{val}})$ (and all algorithmic randomness), $\mathrm{err}_D(h^\star) \leq \epsilon + 2\sqrt{\log(4k/\delta)/(2m_{\mathrm{val}})}$.*

*Proof.* Fix $\epsilon \geq 0$ and $\delta \in (0,1)$. Throughout the proof we condition on the (random) training set $S_{\mathrm{tr}}$ and use that, by assumption, all $k$ trials produce accepted hypotheses using only $S_{\mathrm{tr}}$. Hence, conditioned on $S_{\mathrm{tr}}$, the hypotheses $(h_t)_{t=1}^k$ are i.i.d. draws from the (accepted) proposal distribution $q(\cdot \mid S_{\mathrm{tr}})$ and are independent of $S_{\mathrm{val}}$. (Here $|S_{\mathrm{tr}}| = m_{\mathrm{tr}}$ and $|S_{\mathrm{val}}| = m_{\mathrm{val}}$.)

**Step 1: a near-$\epsilon$ candidate appears with high probability.** Define the set of $\epsilon$-good hypotheses

$$\mathcal{G}_\epsilon := \{h : \mathcal{X} \to \{\pm 1\} \ : \ \mathrm{err}_D(h) \leq \epsilon\}.$$

By definition,

$$p_\epsilon(S_{\mathrm{tr}}) = \Pr_{h \sim q(\cdot | S_{\mathrm{tr}})}[h \in \mathcal{G}_\epsilon].$$

Let $E_{\mathrm{hit}}$ be the event that at least one of the $k$ candidates is $\epsilon$-good:

$$E_{\mathrm{hit}} := \left\{ \exists t \in [k] \text{ s.t. } h_t \in \mathcal{G}_\epsilon \right\}.$$

Conditioned on $S_{\mathrm{tr}}$, since the $h_t$ are i.i.d. from $q(\cdot \mid S_{\mathrm{tr}})$,

$$\Pr\left(E_{\mathrm{hit}}^c \mid S_{\mathrm{tr}}\right) = \Pr\left(\forall t \in [k], \ h_t \notin \mathcal{G}_\epsilon \ \middle| \ S_{\mathrm{tr}}\right) = (1 - p_\epsilon(S_{\mathrm{tr}}))^k.$$

Assume $k \geq \left\lceil \frac{\log(\delta/2)}{\log(1 - p_\epsilon(S_{\mathrm{tr}}))} \right\rceil$. Since $\log(1 - p_\epsilon(S_{\mathrm{tr}})) \leq 0$, this implies $(1 - p_\epsilon(S_{\mathrm{tr}}))^k \leq \delta/2$, and therefore

$$\Pr\left(E_{\mathrm{hit}} \mid S_{\mathrm{tr}}\right) \geq 1 - \delta/2. \tag{2}$$

**Step 2: uniform validation-to-population deviation over $k$ hypotheses.** For each fixed hypothesis $h$, by Hoeffding's inequality applied to the Bernoulli losses $\mathbf{1}\{h(x'_j) \neq y(x'_j)\}$ on $S_{\mathrm{val}}$ with $|S_{\mathrm{val}}| = m_{\mathrm{val}}$,

$$\Pr\left(\left|\mathrm{err}_{S_{\mathrm{val}}}(h) - \mathrm{err}_D(h)\right| > \alpha\right) \leq 2e^{-2m_{\mathrm{val}}\alpha^2}.$$

Conditioned on $S_{\mathrm{tr}}$, the hypotheses $h_1, \ldots, h_k$ are independent of $S_{\mathrm{val}}$, so we can apply the above bound to each $h_t$ and union bound over $t \in [k]$:

$$\Pr\left(\max_{t \in [k]} \left|\mathrm{err}_{S_{\mathrm{val}}}(h_t) - \mathrm{err}_D(h_t)\right| > \alpha \ \middle| \ S_{\mathrm{tr}}\right) \leq \sum_{t=1}^k 2e^{-2m_{\mathrm{val}}\alpha^2} = 2k\, e^{-2m_{\mathrm{val}}\alpha^2}.$$

Choose

$$\alpha := \sqrt{\frac{\log(4k/\delta)}{2m_{\mathrm{val}}}}.$$

Then $2ke^{-2m_{\mathrm{val}}\alpha^2} = 2ke^{-\log(4k/\delta)} = \delta/2$, hence

$$\Pr\left(\forall t \in [k], \ \left|\mathrm{err}_{S_{\mathrm{val}}}(h_t) - \mathrm{err}_D(h_t)\right| \leq \alpha \ \middle| \ S_{\mathrm{tr}}\right) \geq 1 - \delta/2. \tag{3}$$

Let $E_{\mathrm{unif}}$ denote the event inside (3).

**Step 3: conclude the bound on $\mathrm{err}_D(h^\star)$.** On the intersection $E_{\mathrm{hit}} \cap E_{\mathrm{unif}}$, pick an index $t_{\mathrm{good}} \in [k]$ such that $h_{t_{\mathrm{good}}} \in \mathcal{G}_\epsilon$, so $\mathrm{err}_D(h_{t_{\mathrm{good}}}) \leq \epsilon$. Also, on $E_{\mathrm{unif}}$ we have for all $t$,

$$\mathrm{err}_{S_{\mathrm{val}}}(h_t) \leq \mathrm{err}_D(h_t) + \alpha \qquad \text{and} \qquad \mathrm{err}_D(h_t) \leq \mathrm{err}_{S_{\mathrm{val}}}(h_t) + \alpha.$$

Since $h^\star \in \arg\min_{t \in [k]} \mathrm{err}_{S_{\mathrm{val}}}(h_t)$,

$$\mathrm{err}_{S_{\mathrm{val}}}(h^\star) \leq \mathrm{err}_{S_{\mathrm{val}}}(h_{t_{\mathrm{good}}}) \leq \mathrm{err}_D(h_{t_{\mathrm{good}}}) + \alpha \leq \epsilon + \alpha.$$

Applying the other side of the uniform deviation bound to $h^\star$ yields

$$\mathrm{err}_D(h^\star) \leq \mathrm{err}_{S_{\mathrm{val}}}(h^\star) + \alpha \leq \epsilon + 2\alpha = \epsilon + 2\sqrt{\frac{\log(4k/\delta)}{2m_{\mathrm{val}}}}.$$

**Step 4: probability of the good event.** By (2) and (3) and a union bound (still conditioned on $S_{\mathrm{tr}}$),

$$\Pr\big(E_{\mathrm{hit}} \cap E_{\mathrm{unif}} \mid S_{\mathrm{tr}}\big) \;\geq\; 1 - \delta/2 - \delta/2 \;=\; 1 - \delta.$$

Removing the conditioning (i.e., averaging over $S_{\mathrm{tr}}$) preserves the same lower bound, so with probability at least $1 - \delta$ over $(S_{\mathrm{tr}}, S_{\mathrm{val}})$ and all algorithmic randomness,

$$\mathrm{err}_D(h^\star) \;\leq\; \epsilon + 2\sqrt{\frac{\log(4k/\delta)}{2m_{\mathrm{val}}}}.$$

This is exactly the claimed guarantee. $\qquad\square$

## C. Finite-Precision Mini-Batch SGD as a $1$-STAT$(b)$ Statistical Algorithm

Consider optimizing a parameter vector $\theta \in \Theta \subseteq \mathbb{R}^d$ from i.i.d. data $z \sim D$ via a loss $\ell(\theta, z)$. To model $b$-bit access to gradients in the statistical-query framework, we assume each per-example coordinate gradient is uniformly bounded: there exists $G > 0$ such that for all $\theta \in \Theta$, $j \in [d]$, and $z \in \mathcal{Z}$, $|\partial_j \ell(\theta, z)| \leq G$.

**Coordinate mini-batch SGD.** We consider a simple coordinate version of mini-batch SGD. Fix a batch size $B$ and step sizes $(\eta_t)_{t \geq 1}$. Starting from $\theta_1$, at each iteration $t$ the algorithm selects a coordinate $j_t \in [d]$ to update (the choice may depend on the past randomness and observations). It then draws a fresh mini-batch $z_{t,1}, \ldots, z_{t,B} \overset{\text{i.i.d.}}{\sim} D$ and uses these samples to estimate the $j_t$-th partial derivative by the empirical average

$$\widehat{g}_{t,j_t} := \frac{1}{B} \sum_{i=1}^{B} \partial_{j_t} \ell(\theta_t, z_{t,i}).$$

Finally, it takes a step along that coordinate:

$$\theta_{t+1} = \theta_t - \eta_t \, \widehat{g}_{t,j_t} \, e_{j_t}.$$

**Finite-precision model.** To obtain sample/iteration lower bounds, we analyze a finite-precision abstraction of coordinate mini-batch SGD. The point is that, in many implementations, each per-example coordinate gradient is effectively represented with only a small number of bits. This finite-precision view is also convenient for connecting SGD to the statistical-algorithm framework (Reyzin, 2020).

Fix an integer $b \geq 1$ and set $M := 2^b$. Partition the interval $[-G, G]$ into $M$ equal bins of width $\Delta := \frac{2G}{M}$. For any $u \in [-G, G]$, define the (clipped) bin index

$$\mathrm{idx}(u) \;:=\; \min\Big\{M - 1, \; \big\lfloor (u + G)/\Delta \big\rfloor\Big\} \in \{0, 1, \ldots, M - 1\},$$

and map $u$ to the midpoint of its bin:

$$Q_b(u) \;:=\; -G + \Big(\mathrm{idx}(u) + \tfrac{1}{2}\Big)\Delta, \qquad u \in [-G, G].$$

This is a deterministic $b$-bit quantizer with worst-case error at most half a bin: for all $u \in [-G, G]$,

$$|Q_b(u) - u| \leq \Delta/2 = G/2^b. \tag{4}$$

**Definition C.1** ($b$-bit coordinate mini-batch SGD). The $b$-bit variant of coordinate mini-batch SGD replaces each per-example coordinate gradient $\partial_{j_t} \ell(\theta_t^{(b)}, z_{t,i})$ by its quantized value $Q_b(\partial_{j_t} \ell(\theta_t^{(b)}, z_{t,i}))$. At iteration $t$ it forms the quantized mini-batch estimate

$$\widehat{g}_{t,j_t}^{(b)} := \frac{1}{B} \sum_{i=1}^{B} Q_b\big(\partial_{j_t} \ell(\theta_t^{(b)}, z_{t,i})\big),$$

and updates only coordinate $j_t$:

$$\theta_{t+1}^{(b)} = \theta_t^{(b)} - \eta_t \, \widehat{g}_{t,j_t}^{(b)} \, e_{j_t}.$$

Before turning to lower bounds, we introduce a simple stability estimate for finite-precision updates. Intuitively, the $b$-bit run differs from the full-precision run in two ways: each per-example coordinate gradient is quantized, and the two runs may evaluate gradients at slightly different iterates. Under a Lipschitz assumption on per-example coordinate gradients, the accumulated deviation between the two coupled trajectories scales with the rate of precision $O(2^{-b})$.

**Lemma C.2** (Quantization error telescoping under Lipschitz gradients). *Fix $b \in \mathbb{N}$ and a quantizer $Q_b : [-G, G] \to [-G, G]$ satisfying $|Q_b(x) - x| \leq \frac{G}{2^b}$ for all $x \in [-G, G]$. Assume per-example coordinate gradients are bounded and Lipschitz in parameters: there exists $L > 0$ such that for all $\theta, \theta' \in \Theta$, all $z \in \mathcal{Z}$, and all $j \in [d]$,*

$$|\partial_j \ell(\theta, z)| \leq G, \qquad |\partial_j \ell(\theta, z) - \partial_j \ell(\theta', z)| \leq L\|\theta - \theta'\|_2. \tag{5}$$

*At iteration $t$, define the (unquantized) mini-batch coordinate gradient*

$$\widehat{g}_{t,j_t} := \frac{1}{B} \sum_{i=1}^{B} \partial_{j_t} \ell(\theta_t, z_{t,i}) \in [-G, G],$$

*and the $b$-bit mini-batch coordinate gradient*

$$\widehat{g}_{t,j_t}^{(b)} := \frac{1}{B} \sum_{i=1}^{B} Q_b\big(\partial_{j_t} \ell(\theta_t^{(b)}, z_{t,i})\big) \in [-G, G].$$

*Consider two coupled runs that use the same initialization $\theta_1^{(b)} = \theta_1$, the same coordinates $(j_t)_{t \geq 1}$, and the same mini-batches $(z_{t,i})$, updated by*

$$\theta_{t+1} = \theta_t - \eta_t \widehat{g}_{t,j_t} e_{j_t}, \qquad \theta_{t+1}^{(b)} = \theta_t^{(b)} - \eta_t \widehat{g}_{t,j_t}^{(b)} e_{j_t}.$$

*Then, for all $T \geq 1$,*

$$\|\theta_{T+1}^{(b)} - \theta_{T+1}\|_2 \leq \frac{G}{2^b} \sum_{t=1}^{T} \eta_t \prod_{r=t+1}^{T} (1 + L\eta_r) \leq \frac{G}{2^b L} \Big(\exp\big(L \sum_{t=1}^{T} \eta_t\big) - 1\Big), \tag{6}$$

*with the convention that the empty product equals $1$ (so the $t = T$ term is $\eta_T$).*

*Proof.* Fix $t$. Write

$$a_{t,i} := \partial_{j_t} \ell(\theta_t^{(b)}, z_{t,i}), \qquad b_{t,i} := \partial_{j_t} \ell(\theta_t, z_{t,i}).$$

By the triangle inequality,

$$\begin{aligned}
\big|\widehat{g}_{t,j_t}^{(b)} - \widehat{g}_{t,j_t}\big| &= \left|\frac{1}{B} \sum_{i=1}^{B} Q_b(a_{t,i}) - \frac{1}{B} \sum_{i=1}^{B} b_{t,i}\right| \\
&\leq \left|\frac{1}{B} \sum_{i=1}^{B} \big(Q_b(a_{t,i}) - a_{t,i}\big)\right| + \left|\frac{1}{B} \sum_{i=1}^{B} (a_{t,i} - b_{t,i})\right| \\
&\leq \frac{1}{B} \sum_{i=1}^{B} |Q_b(a_{t,i}) - a_{t,i}| + \frac{1}{B} \sum_{i=1}^{B} |a_{t,i} - b_{t,i}|.
\end{aligned}$$

By the quantization property, $|Q_b(a_{t,i}) - a_{t,i}| \leq G/2^b$. By the coordinate Lipschitz condition (5),

$$|a_{t,i} - b_{t,i}| = \big|\partial_{j_t} \ell(\theta_t^{(b)}, z_{t,i}) - \partial_{j_t} \ell(\theta_t, z_{t,i})\big| \leq L\|\theta_t^{(b)} - \theta_t\|_2 \quad \text{for each } i.$$

Therefore,

$$\big|\widehat{g}_{t,j_t}^{(b)} - \widehat{g}_{t,j_t}\big| \leq \frac{G}{2^b} + L\|\theta_t^{(b)} - \theta_t\|_2. \tag{7}$$

Now subtract the updates:

$$\theta_{t+1}^{(b)} - \theta_{t+1} = (\theta_t^{(b)} - \theta_t) - \eta_t(\widehat{g}_{t,j_t}^{(b)} - \widehat{g}_{t,j_t})e_{j_t}.$$

Taking $\ell_2$ norms and using $\|e_{j_t}\|_2 = 1$ gives

$$\|\theta_{t+1}^{(b)} - \theta_{t+1}\|_2 \le \|\theta_t^{(b)} - \theta_t\|_2 + \eta_t |\widehat{g}_{t,j_t}^{(b)} - \widehat{g}_{t,j_t}|.$$

Plugging (7) yields the recursion

$$\Delta_{t+1} \le (1 + L\eta_t)\Delta_t + \eta_t \frac{G}{2^b}, \qquad \text{where } \Delta_t := \|\theta_t^{(b)} - \theta_t\|_2.$$

Assuming $\theta_1^{(b)} = \theta_1$ so that $\Delta_1 = 0$, unrolling gives

$$\Delta_{T+1} \le \frac{G}{2^b} \sum_{t=1}^{T} \eta_t \prod_{r=t+1}^{T} (1 + L\eta_r),$$

which is the first bound in (6).

For the second bound, use $1 + u \le e^u$ to obtain

$$\prod_{r=t+1}^{T} (1 + L\eta_r) \le \exp\Big(L \sum_{r=t+1}^{T} \eta_r\Big).$$

Define $S_t := \sum_{r=t}^{T} \eta_r$ and $A_t := e^{LS_t}$, so $A_t = e^{L\eta_t} A_{t+1}$ and $A_{T+1} = 1$. Then

$$A_t - A_{t+1} = (e^{L\eta_t} - 1)A_{t+1} \ge L\eta_t A_{t+1} \implies \eta_t A_{t+1} \le \frac{1}{L}(A_t - A_{t+1}).$$

Summing this inequality over $t = 1, \ldots, T$ yields

$$\sum_{t=1}^{T} \eta_t \exp\Big(L \sum_{r=t+1}^{T} \eta_r\Big) = \sum_{t=1}^{T} \eta_t A_{t+1} \le \frac{1}{L} \sum_{t=1}^{T}(A_t - A_{t+1}) = \frac{1}{L}(A_1 - A_{T+1}) = \frac{1}{L}\Big(e^{L\sum_{t=1}^{T} \eta_t} - 1\Big).$$

Therefore,

$$\Delta_{T+1} \le \frac{G}{2^b L}\Big(\exp\big(L \sum_{t=1}^{T} \eta_t\big) - 1\Big),$$

establishing (6). $\qquad\square$

## C.1. $b$-Bit Coordinate Mini-Batch SGD as a 1-STAT$(b)$ Algorithm

A 1-STAT$(b)$ query is a vector of $b$ Boolean functions $g = (g_1, \ldots, g_b)$ with $g_k : \mathcal{Z} \to \{0,1\}$. The oracle draws a fresh $z \sim D$ and returns $(g_1(z), \ldots, g_b(z)) \in \{0,1\}^b$.

**Encoding the quantized gradient as a 1-STAT$(b)$ answer.** Fix $(\theta, j)$. Recall that the quantizer $Q_b$ defined above maps $[-G, G]$ to exactly $M := 2^b$ midpoints, indexed by $\{0, \ldots, M-1\}$. For a sample $z \in \mathcal{Z}$, define

$$\mathrm{idx}_{\theta,j}(z) := \mathrm{idx}\big(\partial_j \ell(\theta, z)\big) \in \{0, 1, \ldots, M-1\}.$$

For each $r \in \{1, \ldots, b\}$, define the Boolean function $g_r^{\theta,j} : \mathcal{Z} \to \{0,1\}$ as the $r$-th bit (under a fixed convention) of the binary representation of $\mathrm{idx}_{\theta,j}(z)$:

$$g_r^{\theta,j}(z) := \mathrm{bit}_r\big(\mathrm{idx}_{\theta,j}(z)\big).$$

Then one call to 1-STAT$(b)$ on $(g_1^{\theta,j}, \ldots, g_b^{\theta,j})$ returns the $b$ bits encoding $\mathrm{idx}_{\theta,j}(z)$ for a fresh sample $z \sim D$, from which the algorithm reconstructs

$$Q_b\big(\partial_j \ell(\theta, z)\big) = -G + \Big(\mathrm{idx}_{\theta,j}(z) + \tfrac{1}{2}\Big)\Delta.$$

**Lemma C.3** ($b$-bit mini-batch SGD is a 1-STAT$(b)$ statistical algorithm). *A $T$-iteration run of $b$-bit coordinate mini-batch SGD (Definition C.1) with batch size $B$ can be implemented using exactly $q = TB$ calls to 1-STAT$(b)$, plus internal computation.*

*Proof.* At iteration $t$, for each $i = 1, \ldots, B$ the algorithm needs the quantized value $Q_b(\partial_{j_t} \ell(\theta_t^{(b)}, z_{t,i}))$ for a fresh $z_{t,i} \sim D$. Fix $\theta = \theta_t^{(b)}$ and $j = j_t$. Using the encoding above, one call to 1-STAT($b$) on $(g_1^{\theta,j}, \ldots, g_b^{\theta,j})$ returns the $b$ bits encoding $\mathrm{idx}_{\theta,j}(z_{t,i})$, hence allows reconstruction of $Q_b(\partial_j \ell(\theta, z_{t,i}))$. Repeating this for $i = 1, \ldots, B$ yields the $B$ quantized per-example gradients, which can be averaged to obtain $\widehat{g}_{t,j_t}^{(b)}$ and then used to update $\theta_{t+1}^{(b)}$. Over $T$ iterations this uses exactly $TB$ oracle calls. $\qquad\square$

**Lemma C.4** (SDA for planted $k$-parity testing)**.** *Let $n \geq 1$, $k \in \{1, \ldots, n\}$, and let*

$$\mathcal{X} = \{0,1\}^n, \qquad \mathcal{Y} = \{\pm 1\}, \qquad \mathcal{S}_k := \{s \in \{0,1\}^n : \|s\|_0 = k\}, \qquad N := |\mathcal{S}_k| = \binom{n}{k}.$$

*For each $s \in \mathcal{S}_k$, let $D_s$ be the realizable distribution on $\mathcal{X} \times \mathcal{Y}$ given by*

$$x \sim \mathrm{Unif}(\mathcal{X}), \qquad y = f_s(x) := (-1)^{\langle s, x \rangle}, \qquad \text{where } \langle s, x \rangle := \sum_{i=1}^n s_i x_i \pmod 2.$$

*Let $D_0$ be the null distribution where $x \sim \mathrm{Unif}(\mathcal{X})$ and $y \sim \mathrm{Unif}(\mathcal{Y})$ independently. Let $\chi_{D_0}(\cdot, \cdot)$ and $\rho(\cdot, D_0)$ be as in Definitions 27–28 of* Reyzin (2020) *(equivalently,* Feldman et al. (2018)*).*

*Then for every nonempty $\mathcal{D}' \subseteq \{D_s : s \in \mathcal{S}_k\}$ with $|\mathcal{D}'| = m$, we have*

$$\rho(\mathcal{D}', D_0) = \frac{1}{m}.$$

*(Here $\rho$ includes diagonal terms, i.e., $D_1, D_2$ are drawn independently and uniformly from $\mathcal{D}'$, so $\Pr[D_1 = D_2] = 1/|\mathcal{D}'|$.)*

*Consequently, for the promise testing problem $Z$ (distinguish $D_0$ from some $D_s$), define $\mathrm{SDA}_{\mathbb{Z}}(Z, \bar\gamma)$ to be the integer-valued statistical dimension with average correlation, namely the largest integer $d \geq 1$ such that for every $\mathcal{D}' \subseteq \{D_s : s \in \mathcal{S}_k\}$ with $|\mathcal{D}'| \geq N/d$, we have $\rho(\mathcal{D}', D_0) \leq \bar\gamma$; if no such integer exists, define $\mathrm{SDA}_{\mathbb{Z}}(Z, \bar\gamma) := 0$. Then for every $\bar\gamma \in (0,1)$,*

$$\mathrm{SDA}_{\mathbb{Z}}(Z, \bar\gamma) = \begin{cases} \Theta(\bar\gamma N), & \text{if } \bar\gamma \geq 1/N, \\ 0, & \text{if } \bar\gamma < 1/N. \end{cases}$$

*In particular, when $\bar\gamma < 1/N$, Theorem 30 is vacuous in this regime (and in particular, it does not imply any $\omega(1)$ query lower bound).*

*Remark* C.5 (Real-valued SDA and the $\bar\gamma < 1/N$ regime). Definition 29 in Reyzin (2020) defines $\mathrm{SDA}(Z, \bar\gamma)$ as the *largest real* $d > 0$ such that for every $\mathcal{D}' \subseteq \{D_s : s \in \mathcal{S}_k\}$ with $|\mathcal{D}'| \geq N/d$ we have $\rho(\mathcal{D}', D_0) \leq \bar\gamma$. When a maximum need not be attained, we use the standard extension

$$\mathrm{SDA}^{\mathrm{sup}}(Z, \bar\gamma) := \sup\left\{d > 0 : \forall \mathcal{D}' \subseteq \{D_s\} \text{ with } |\mathcal{D}'| \geq N/d, \ \rho(\mathcal{D}', D_0) \leq \bar\gamma\right\}.$$

In our setting, if $\bar\gamma < 1/N$ then the feasible set equals $(0, 1)$ (all $d \in (0, 1)$ make the condition vacuous, while no $d \geq 1$ is feasible), and hence $\mathrm{SDA}^{\mathrm{sup}}(Z, \bar\gamma) = 1$. Therefore Theorem 30 yields at most an $O(1)$ (in particular, not $\omega(1)$) query lower bound in this regime.

*Proof.* Fix $s, s' \in \mathcal{S}_k$. For $(x, y) \in \mathcal{X} \times \mathcal{Y}$, under the null distribution $D_0$ we have $x \sim \mathrm{Unif}(\mathcal{X})$ and $y \sim \mathrm{Unif}(\mathcal{Y})$ independent, hence

$$D_0(x, y) = 2^{-(n+1)}.$$

Under $D_s$, we have $x \sim \mathrm{Unif}(\mathcal{X})$ and $y = f_s(x)$ deterministically, so

$$D_s(x, y) = 2^{-n} \cdot \mathbf{1}\{y = f_s(x)\}.$$

Therefore

$$\frac{D_s(x, y)}{D_0(x, y)} = 2 \cdot \mathbf{1}\{y = f_s(x)\}, \qquad \text{and hence} \qquad \frac{D_s}{D_0} - 1 = \begin{cases} 1, & y = f_s(x), \\ -1, & y = -f_s(x), \end{cases} = y f_s(x).$$

By Definition 27 (pairwise correlation),

$$\chi_{D_0}(D_s, D_{s'}) = \mathbb{E}_{(x,y)\sim D_0}\left[\left(\frac{D_s}{D_0}-1\right)\left(\frac{D_{s'}}{D_0}-1\right)\right]$$
$$= \mathbb{E}_{(x,y)\sim D_0}\left[y f_s(x) \cdot y f_{s'}(x)\right] = \mathbb{E}_{x\sim\mathrm{Unif}(\mathcal{X})}\left[f_s(x) f_{s'}(x)\right].$$

Moreover,

$$f_s(x) f_{s'}(x) = (-1)^{\langle s,x\rangle + \langle s',x\rangle} = (-1)^{\langle s\oplus s',x\rangle},$$

where $\oplus$ is bitwise XOR. Since $x$ is uniform on $\{0,1\}^n$,

$$\mathbb{E}_x[(-1)^{\langle t,x\rangle}] = \begin{cases} 1, & t = 0, \\ 0, & t \neq 0, \end{cases}$$

so $\chi_{D_0}(D_s, D_{s'}) = 1$ if $s = s'$ and $0$ otherwise.

Now let $\mathcal{D}' \subseteq \{D_s : s \in \mathcal{S}_k\}$ be any nonempty subset of size $m$. Using Definition 28 (average correlation) and the above orthogonality, only the $m$ diagonal terms contribute:

$$\rho(\mathcal{D}', D_0) = \frac{1}{m^2}\sum_{D_1,D_2\in\mathcal{D}'}\chi_{D_0}(D_1, D_2) = \frac{1}{m^2}\cdot m = \frac{1}{m}.$$

We now compute $\mathrm{SDA}_{\mathbb{Z}}(Z,\bar\gamma)$. Fix an integer $d \geq 1$ and write

$$m_0 := \left\lceil\frac{N}{d}\right\rceil.$$

Since $\rho(\mathcal{D}', D_0) = 1/|\mathcal{D}'|$ is decreasing in $|\mathcal{D}'|$, the worst case among $|\mathcal{D}'| \geq N/d$ occurs at the smallest allowed size, namely $|\mathcal{D}'| = m_0$. Thus the defining condition for $\mathrm{SDA}_{\mathbb{Z}}(Z,\bar\gamma)$ is equivalent to

$$\frac{1}{m_0} \leq \bar\gamma.$$

**Regime 1:** $\bar\gamma < 1/N$. For every integer $d \geq 1$ we have $m_0 = \lceil N/d\rceil \leq N$, hence $1/m_0 \geq 1/N > \bar\gamma$. So no integer $d \geq 1$ is feasible and $\mathrm{SDA}_{\mathbb{Z}}(Z,\bar\gamma) = 0$.

**Regime 2:** $\bar\gamma \geq 1/N$. We show $\mathrm{SDA}_{\mathbb{Z}}(Z,\bar\gamma) = \Theta(\bar\gamma N)$.

**Lower bound.** Let $d := \max\{1, \lfloor\bar\gamma N/2\rfloor\}$. If $d = 1$, then $m_0 = \lceil N\rceil = N$ and $1/m_0 = 1/N \leq \bar\gamma$, so $d$ is feasible. Otherwise $d = \lfloor\bar\gamma N/2\rfloor \geq 1$, so $d \leq \bar\gamma N/2$ and hence $N/d \geq 2/\bar\gamma$. Therefore

$$m_0 = \left\lceil\frac{N}{d}\right\rceil \geq \left\lceil\frac{2}{\bar\gamma}\right\rceil \geq \frac{1}{\bar\gamma},$$

which implies $1/m_0 \leq \bar\gamma$. Thus $d$ is feasible and $\mathrm{SDA}_{\mathbb{Z}}(Z,\bar\gamma) \geq \Omega(\bar\gamma N)$.

**Upper bound.** If $\bar\gamma \geq 1/2$, then $\bar\gamma N = \Theta(N)$ and trivially $\mathrm{SDA}_{\mathbb{Z}}(Z,\bar\gamma) \leq N$, so $\mathrm{SDA}_{\mathbb{Z}}(Z,\bar\gamma) = \Theta(\bar\gamma N)$.

Assume $\bar\gamma < 1/2$ and let $d > 4\bar\gamma N$. Then $N/d < 1/(4\bar\gamma)$ and hence

$$m_0 = \left\lceil\frac{N}{d}\right\rceil \leq \left\lceil\frac{1}{4\bar\gamma}\right\rceil < \frac{1}{\bar\gamma},$$

where the last inequality uses $\bar\gamma < 1/2$. Therefore $1/m_0 > \bar\gamma$, so the condition fails and no such $d$ is feasible. Thus $\mathrm{SDA}_{\mathbb{Z}}(Z,\bar\gamma) \leq O(\bar\gamma N)$ for $\bar\gamma < 1/2$.

Combining the lower and upper bounds yields $\mathrm{SDA}_{\mathbb{Z}}(Z,\bar\gamma) = \Theta(\bar\gamma N)$ for $\bar\gamma \geq 1/N$. $\qquad\square$

**Theorem C.6** (Iteration/sample lower bound for $b$-bit mini-batch coordinate SGD on $k$-parity learning). *Let $n \geq 1$ and $k \in \{1, \ldots, n\}$, and let $N = \binom{n}{k}$. For each $k$-sparse $s \in \mathcal{S}_k \subseteq \{0,1\}^n$, let $D_s$ be the realizable distribution over $\mathcal{X} \times \{\pm 1\}$ given by*

$$x \sim \mathrm{Unif}(\{0,1\}^n), \qquad y = f_s(x) = (-1)^{\langle s, x \rangle}.$$

*Consider a procedure that runs $T$ iterations of $b$-bit coordinate mini-batch SGD with batch size $B$ (Definition C.1), and whose interaction with fresh examples $(x, y) \sim D_s$ is only through the quantized per-example coordinate gradients (equivalently, through 1-STAT($b$) access as in Lemma C.3). Let the procedure output a hypothesis $h : \{0,1\}^n \to \{\pm 1\}$.*

*Assume that for every $s \in \mathcal{S}_k$ the procedure achieves nontrivial population error with probability at least $\beta \geq 5/6$, namely*

$$\Pr\left[ \mathrm{err}_{D_s}(h) \leq \frac{1}{4} \right] \geq \beta, \qquad where \qquad \mathrm{err}_{D_s}(h) := \Pr_{(x,y) \sim D_s}[h(x) \neq y].$$

*Let $q := TB$ be the total number of fresh examples used (equivalently, the number of 1-STAT($b$) queries). Then necessarily*

$$q \, 2^b = \Omega\left( \sqrt{N} \right),$$

*and hence*

$$q = \Omega\left( \frac{\sqrt{N}}{2^b} \right), \qquad and\ consequently \qquad T = \Omega\left( \frac{\sqrt{N}}{B \, 2^b} \right).$$

*Proof.* **Step 1: $b$-bit SGD is a 1-STAT($b$) algorithm.** By Lemma C.3, a $T$-iteration run with batch size $B$ uses exactly

$$q := TB$$

calls to 1-STAT($b$).

**Step 2: Simulate 1-STAT($b$) by VSTAT.** Apply Theorem B.4 in (Feldman, 2017) with a fixed constant $\delta := 1/100$. This yields an algorithm $\mathcal{A}'$ that succeeds with probability at least $\beta - \delta > 2/3$ and uses at most

$$Q = O(q \, 2^b)$$

queries to VSTAT($t$) with

$$t = \Theta\left( \frac{q \, 2^b}{\delta^2} \right) = \Theta(q \, 2^b),$$

where the hidden constants may depend on $\delta$ but $\delta$ is fixed. In particular, by choosing $\delta$ sufficiently small as an absolute constant, we may assume that whenever $q 2^b \geq 1$ the corresponding parameter $t$ satisfies $t \geq 64$.

**Step 3: Learning implies testing using one additional VSTAT($t$) query.** Define the promise testing problem $Z$: given oracle access to an unknown distribution $D$ that is either $D_0$ (where $x \sim \mathrm{Unif}(\{0,1\}^n)$ and $y \sim \mathrm{Unif}(\{\pm 1\})$ independent) or $D_s$ for some unknown $s \in \mathcal{S}_k$, output PLANTED if $D = D_s$ and NULL if $D = D_0$.

Fix any $s \in \mathcal{S}_k$. Condition on the event that the learner outputs $h$ with $\mathrm{err}_{D_s}(h) \leq 1/4$, and consider the bounded query

$$\phi_h(x, y) := \frac{1 + h(x)y}{2} \in [0, 1].$$

Under $D_s$,

$$\mathbb{E}_{(x,y) \sim D_s}[\phi_h(x, y)] = \Pr[h(x) = y] = 1 - \mathrm{err}_{D_s}(h) \geq \frac{3}{4}.$$

Under $D_0$, since $y$ is independent uniform and $h(x) \in \{\pm 1\}$,

$$\mathbb{E}_{(x,y) \sim D_0}[h(x)y] = 0 \qquad \Rightarrow \qquad \mathbb{E}_{(x,y) \sim D_0}[\phi_h(x, y)] = \frac{1}{2}.$$

Now make one additional VSTAT($t$) query with $\phi_h$ and let $v$ be the oracle's response. Output PLANTED iff $v \geq 5/8$, else output NULL.

For the standard $\text{VSTAT}(t)$ oracle (Feldman et al., 2018), for any $[0, 1]$-valued query with mean $p$ the returned value satisfies

$$|v - p| \leq \max\left\{\frac{1}{t}, \sqrt{\frac{p(1-p)}{t}}\right\}.$$

Since $t \geq 64$, this error is at most $1/16$ both at $p = 1/2$ and at $p = 3/4$, so the threshold $5/8$ separates the two cases:

$$D = D_0 : \ v \leq \frac{1}{2} + \frac{1}{16} = \frac{9}{16} < \frac{5}{8}, \qquad D = D_s : \ v \geq \frac{3}{4} - \frac{1}{16} = \frac{11}{16} > \frac{5}{8}.$$

Thus, whenever $\text{err}_{D_s}(h) \leq 1/4$, this extra query yields a correct test.

Since $\mathcal{A}'$ outputs such an $h$ with probability at least $2/3$ on $D_s$, the composed tester succeeds with probability at least $2/3$ as well. It uses at most $Q + 1 = O(q2^b)$ queries to the *same* oracle $\text{VSTAT}(t)$.

**Step 4: Apply the $\text{VSTAT}$ lower bound.** Apply Theorem 30 of Reyzin (2020) to the promise testing problem $Z$. That theorem is parameterized by an accuracy $\bar{\gamma} > 0$ and assumes oracle access to $\text{VSTAT}(\lceil c_1/\bar{\gamma}\rceil)$ for a universal constant $c_1 > 0$.

Fix another universal constant $c_2 > 0$ and set

$$\bar{\gamma} := \frac{c_2}{t}.$$

Then

$$\left\lceil\frac{c_1}{\bar{\gamma}}\right\rceil = \left\lceil\frac{c_1}{c_2}t\right\rceil = \Theta(t).$$

Since the $\text{VSTAT}(t)$ accuracy guarantee is monotone in $t$ (larger $t$ means smaller allowed error), any response admissible for $\text{VSTAT}(t)$ is also admissible for $\text{VSTAT}(t')$ for every $t' \leq t$. Therefore, by choosing the constant $c_2 > 0$ (in the definition $\bar{\gamma} = c_2/t$) sufficiently small so that

$$\left\lceil\frac{c_1}{\bar{\gamma}}\right\rceil = \left\lceil\frac{c_1}{c_2}t\right\rceil \leq t,$$

we ensure that oracle access to $\text{VSTAT}(t)$ suffices to instantiate Theorem 30 at accuracy $\bar{\gamma}$ (up to constant factors in the parameterization).

Let $d := \text{SDA}(Z, \bar{\gamma})$. By Lemma C.4,

$$d = \begin{cases} \Theta(\bar{\gamma}N) = \Theta(N/t), & \text{if } \bar{\gamma}N \geq 1 \ (\text{i.e. } t \leq c_2N), \\ \Theta(1), & \text{if } \bar{\gamma}N < 1 \ (\text{i.e. } t > c_2N). \end{cases}$$

Theorem 30 implies that any algorithm that solves $Z$ with probability at least $2/3$ using only $\text{VSTAT}(t)$ access must make at least $\Omega(d)$ oracle queries.

We have such a tester using $O(q2^b)$ queries. Therefore:

- If $t \leq c_2N$, then $d = \Theta(N/t)$ and hence

$$q2^b = \Omega(N/t).$$

  With $t = \Theta(q2^b)$ from Step 2 this gives

$$q2^b = \Omega\left(\frac{N}{q2^b}\right) \quad \implies \quad (q2^b)^2 = \Omega(N) \quad \implies \quad q = \Omega\left(\frac{\sqrt{N}}{2^b}\right).$$

- If $t > c_2N$, then $t = \Theta(q2^b)$ implies $q2^b = \Omega(N)$, which in particular implies $q2^b = \Omega(\sqrt{N})$ and hence $q = \Omega(\sqrt{N}/2^b)$ as well.

Finally, since $q = TB$, the lower bound on $q$ implies $T = \Omega\left(\frac{\sqrt{N}}{B\,2^b}\right)$. $\qquad\square$

