# OpenReview forum: "LLM Priors for ERM over Programs"
_ICML.cc/2026/Conference — ICML 2026 regular_

### Official Review · Reviewer_mMwn · 2026-03-08

**Soundness:** 2
**Presentation:** 2
**Significance:** 3
**Originality:** 2
**Overall Recommendation:** 3
**Confidence:** 3

**Summary:**

This paper studies small-data learning problems where the target rule is essentially algorithmic. Instead of exhaustively searching over all short programs or relying on SGD to discover the rule, the authors use a pretrained LLM to propose a few candidate programs, execute them, and pick the one that works best on validation data. They show that this works well on several synthetic rule-learning tasks and generalizes to longer inputs better than fine-tuning, in-context learning, SGD-trained transformers, and standard baselines. The paper’s main takeaway is that LLMs may be useful not just as predictors, but as proposal mechanisms for program search.

**Compliance With Llm Reviewing Policy:**

Affirmed.

**Key Questions For Authors:**

- How sensitive is LLM-PV to proposer quality: The theory and empirical success both seem to hinge on the proposer assigning sufficient mass to good candidate programs. Can the authors report sensitivity to the number of samples, and ablations across proposer models and prompt variants?

- Why are there no direct comparisons to stronger synthesis-oriented or LLM-guided search baselines?

- How should the reader interpret the theory relative to the experimental section?
Is the intention that the lower-bound discussion is mainly motivational, or do the authors believe it directly explains the failures?

- How broad do the authors believe the intended scope of the method is? Do the authors view this primarily as a method for program-like latent rules, or as a more general supervised learning paradigm?

**Limitations:**

yes

**Strengths And Weaknesses:**

I liked the core idea of the paper: If the target rule has a short programmatic description, ERM over programs is statistically attractive, but brute-force search does not scale; meanwhile SGD is easy to run but often fails to recover these structured rules. The proposed fix, LLM-PV, is neat: use the LLM to propose a small number of candidate programs, run them, and keep the one with the lowest validation error. That gives the method a real interpretability advantage over direct prediction, since the final output is executable code rather than just a labeler. The paper also makes this concrete by showing candidate traces and an explicit reasoning path for primality.

The empirical results on the synthetic tasks are strong. The main setup uses 200 i.i.d. examples split into 100 train and 100 validation, with a 10,000-example test set. On the length-generalization experiments, LLM-PV stays strong out to length 100 while SGD, fine-tuning, and ICL mostly collapse toward chance. The “100k examples still do not rescue SGD” result is also notable: BLOOM-75M gets perfect train accuracy on 10-P and CAP but test accuracy stays near chance. Those are meaningful findings in the regime the paper studies.

I found the framing interesting, but I am less convinced by the novelty than the paper’s tone suggests. The ingredients are close to PBE / inductive synthesis / propose-and-verify: search a discrete program space, verify candidates by execution, and use the LLM as a search bias. The paper’s own related-work section more or less says this; its main distinction is that it repurposes this setup for i.i.d. learning with held-out validation instead of plain consistency with examples. That is a real contribution, but it feels more like a reframing plus strong empirical evidence than a fundamentally new algorithmic idea. The paper compares mainly against direct prediction baselines: SGD-trained classifiers, fine-tuning, ICL, and classical ML but not against LLM-guided synthesis systems.

HYSYNTH: Context-Free LLM Approximation for Guiding Program Synthesis: probably the closest omission; it is also an LLM-guided program synthesis method that combines neural proposals with symbolic search: https://arxiv.org/pdf/2405.15880

LLM-Guided Compositional Program Synthesis https://arxiv.org/pdf/2503.15540

I also think the evaluation is narrower than the paper sometimes suggests. The tasks are synthetic algorithmic benchmarks where the true rule really is a compact symbolic program, and the prompt directly asks the model to synthesize concise Python code. In that setting, the method is a very good fit.

---

> ### Author Rebuttal · Authors · 2026-03-31
>
> We thank the reviewer for their thoughtful and constructive feedback.
>
> New experiments: https://anonymous.4open.science/r/LLM_PV-C04C/experiments/
>
> For a summary see the first comment to reviewer 1nTN.
>
> > Reviewer: Novelty relative to prior LLM-guided synthesis methods
>
> Response: We agree that our method overlaps conceptually with prior propose-and-verify, PBE, and LLM-guided synthesis work, and we do not claim a new search primitive. Our contribution is to repurpose this paradigm for i.i.d. supervised learning with a generalization objective: candidate programs are proposed by an LLM but selected by held-out validation error rather than consistency with a finite specification alone. The novelty is thus in the learning formulation, evaluation lens, and empirical evidence that this approach can outperform standard direct-prediction baselines when exhaustive ERM over short programs is statistically attractive but computationally infeasible. We also agree that HYSYNTH and related synthesis systems should be discussed explicitly, but they are not drop-in baselines for our benchmark because they rely on domain-specific DSLs or other task-specific symbolic infrastructure, whereas our setting is a Python-level supervised-learning benchmark. We now include a direct comparison to SoTA LLM-guided search method: MLAgentBench, which performs substantially worse than LLM-PV in our setting (exp 3).
>
> > Reviewer: How broad is the evaluation beyond synthetic algorithmic tasks?
>
> Response: We agree that much of the evaluation focuses on algorithmic tasks with compact latent program structure, which are especially well aligned with the method. This is intentional: the paper is designed to test whether PV can recover structured rules from very limited supervision. That said, the evaluation is not limited to synthetic sequence tasks; we also include tabular benchmarks, where LLM-PV is competitive with strong baselines. We will clarify that our claim is not broad coverage of supervised learning, but effectiveness on structured learning problems.
>
> > Reviewer: How sensitive is LLM-PV to proposer quality, sample budget, and prompt/model choices?
>
> Response: We added proposer ablations across multiple models, including Gemini 3.1 Pro, Qwen3-30B-A3B-Thinking, DeepSeek-Coder-33B-Instruct, OLMo-3-32B-Thinking, and GPT-5-Thinking without tool use (exp 1). The pattern is consistent: strong agentic proposers perform well, while weaker or non-agentic proposers degrade substantially; supporting the theoretical dependence on proposal mass over good candidate programs. Regarding sample budget, we ablated the number of samples in Fig. 4. For prompt sensitivity, we added an experiment using 4 randomly generated prompts; performance was broadly similar to our original results (exp 5).
>
> > Reviewer: Why are there no direct comparisons to stronger synthesis-oriented or LLM-guided search baselines?
>
> Response: We added a direct comparison to MLAgentBench, an LLM-guided baseline in which GPT-5-Thinking iteratively refines an ML pipeline to improve performance. In the revised results, MLAgentBench performs substantially worse than LLM-PV, supporting our claim that PV search is more effective than generic pipeline refinement for these tasks (exp 3). More specialized synthesis systems such as HYSYNTH and LLM-Guided Compositional Program Synthesis are also relevant, but they rely on domain-specific DSLs, symbolic execution, or other task-specific synthesis machinery that is not directly available in our Python-level supervised-learning setting. We now clarify this limitation explicitly.
>
> > Reviewer: Interpreting the theory relative to the experiments
>
> Response: Our intention is that the lower-bound discussion is more than motivational. It highlights an SQ-style barrier for these structured rule-learning tasks: aggregate statistical fitting can be poorly matched to recovering the underlying discrete program, whereas explicit proposal-based search is better aligned with the hypothesis class. We do not claim that the experiments literally instantiate the SQ model or that the bound predicts each failure quantitatively; rather, we view it as explaining the broad empirical pattern.
>
> > Reviewer: Intended scope: program-like latent rules or broader supervised learning?
>
> Response: We do not view LLM-PV as a general replacement for supervised deep learning. Its clearest scope is structured learning settings, especially algorithmic tasks, where the goal is to uncover latent structure and synthesize an explicit solution. Program-like latent rules are thus the natural focus of this paper. More broadly, the framework may extend to larger-scale settings, including some tabular regimes, through iterative PV-refine updates, but we view this as a direction rather than an established claim. A further advantage is interpretability: the learned predictor is explicit executable code, and the LLM-guided search process is often easier to inspect than standard black-box training.

---

> > ### Author Rebuttal · Reviewer_mMwn · 2026-04-05
> >
> > The rebuttal addresses several of my concrete questions: it adds proposer/sample/prompt ablations, narrows the intended scope to structured program-like tasks rather than broad supervised learning, and includes a new comparison to MLAgentBench. That said, my core novelty concern is only partially resolved so I would lean toward keeping my score unchanged.

---

> > > ### Author Response · Authors · 2026-04-05
> > >
> > > Thank you for the follow-up. We would like to clarify more sharply why we believe the novelty is more substantial than “reframing plus strong empirical evidence.”
> > >
> > > We agree that the paper does not introduce a new search primitive in the program-synthesis sense. However, that is not the level at which the main novelty lies. The contribution is a new learning formulation: i.i.d. supervised learning over a discrete program class, with generalization as the objective and held-out validation-based model selection over executable hypotheses. This is different from the classical PBE / synthesis setting, where the goal is typically consistency with a finite specification, often within a DSL or with task-specific symbolic machinery.
> > >
> > > That distinction is not cosmetic. It changes both the learning lens and the technical message of the paper. On the theory side, we formalize a concrete compute-sample tension: short-program ERM is statistically attractive but computationally infeasible by enumeration, while gradient-based learning can be computationally convenient yet sample-inefficient on structured SQ-hard families. On the algorithmic side, LLM-PV uses the pretrained LLM only as a proposal prior, while keeping selection grounded in executable verification and held-out error, rather than using the model itself as the predictor or adapting it via finetuning, synthetic data generation, or iterative gradient-style feedback. On the empirical side, the new controls show that the gain is not explained simply by access to a strong frontier model: same-backbone GPT-5 ICL and MLAgentBench with GPT-5 still underperform LLM-PV.
> > >
> > > So our claim is not merely that LLMs can help synthesis. Rather, the paper shows that foundation-model-guided search can itself serve as a viable machine-learning mechanism for ERM over structured hypothesis classes. In that sense, we believe the paper opens a new avenue for ML approaches based on guided search, not only gradient-based adaptation.
> > > We will revise the paper to make this point more explicit and better separated from the narrower claim of algorithmic novelty in the synthesis sense, and we will expand the discussion of HYSYNTH and related work accordingly.
> > >
> > > Finally, we also added new experiments to strengthen the comparisons, including another method AIDE-ML of the same family as MLAgentBench, as well as additional ICL ablations with tool use:
> > >
> > > | Tasks | ICL: GPT 5 Thinking without Tools | ICL: Gemini 3.1 Pro Thinking with Tools | ICL: GPT 5 Thinking with Tools | MLAgentBench with GPT 5 Thinking with Tools | AIDE-ML with GPT 5 Thinking with Tools | LLM-PV: GPT 5 Thinking with Tools |
> > > | :--- | :--- | :--- | :--- | :--- | :--- | :--- |
> > > | FP | 82.0 | 48.0 | 99.0 | 53.6 | 49.5 | **100** |
> > > | FHP | 66.0 | 54.0 | 88.0 | 51.0 | 49.9 | **100** |
> > > | Pattern1 | 55.0 | 83.0 | 51.0 | 33.4 | 50.3 | **100** |
> > > | Pattern2 | 41.0 | 57.0 | 51.0 | 60.9 | 50.9 | **100** |
> > > | 3-P | 47.0 | 50.0 | 75.0 | 32.8 | 49.5 | **100** |
> > > | 10-P | 47.0 | 45.0 | 79.0 | 50.1 | 49.2 | **100** |
> > > | IsPal | 79.0 | 100.0 | 91.0 | 31.6 | 50.0 | **100** |
> > > | Dyck-2 | 57.0 | 51.0 | 59.0 | 52.1 | 50.5 | **100** |
> > > | CAP | 51.0 | 51.0 | 46.0 | 50.0 | 50.2 | **100** |
> > > | isPrime | 55.0 | 52.0 | 62.0 | 32.6 | 55.5 | **100** |
> > > | isPrime+47 | 47.0 | 56.0 | 56.0 | 33.1 | 49.7 | **80.6** |
> > > | CycleCount (seq_len=300) | 43.0 | 50.0 | 51.0 | 49.1 | 50.0 | **96.9** |
> > > | SHA | 44.0 | 45.0 | 46.0 | 49.9 | **50.8** | 50.1 |

---

### Official Review · Reviewer_1nTN · 2026-03-12

**Soundness:** 3
**Presentation:** 4
**Significance:** 3
**Originality:** 4
**Overall Recommendation:** 5
**Confidence:** 3

**Summary:**

This paper proposes a new learning algorithm, LLM-PV, an ERM-based approach for estimating small programs. If the hypothesis class is finite, an ERM-based approach is known to estimate a hypothesis with small error with a small number of training samples. The hypothesis space of fixed programs is finite but prohibitively large, making it difficult to find an ERM hypothesis. The proposed method leverages an LLM to sample high-quality hypotheses. The proposed LLM-PV method generates program samples using an LLM and then selects a hypothesis that minimizes validation error. If an LLM can find high-quality samples, the learning paradigm guarantees that a program with a small number of training samples will converge to a program with a guaranteed error.

**Compliance With Llm Reviewing Policy:**

Affirmed.

**Final Justification:**

I have decided to maintain my initial score. The proposed LLM-PV framework and its underlying theoretical foundation, based on ERM learnability, are both novel and compelling. During the rebuttal phase, the authors successfully addressed my concerns regarding whether the experimental performance was primarily driven by the inherent power of the LLMs.

**Key Questions For Authors:**

Why do the authors use GPT-5-thinking only for the LLM-PV? How would the results change if we used a different model?

**Limitations:**

Yes

**Strengths And Weaknesses:**

**Strenghts**
- The concept of combining a classical ERM-based learning with a finite set of hypotheses with LLM is very interesting.
- Experimental results show the superiority of the proposed approach over a wide range of baseline methods when the number of training samples is small. These results highlight the unique characterization of the proposed approach.
- The paper is very clearly written and easy to follow.

**Weaknesses**

- In experiments, the proposed method is compared with baseline LLM-based methods. Although the experimental results seem promising, the selection of LLMs seems unfair, as only LLM-PV uses the most powerful LLM, GPT-5-thinking.

---

> ### Author Rebuttal · Authors · 2026-03-31
>
> We thank the reviewer for their thoughtful and constructive feedback.
>
> New experiments: https://anonymous.4open.science/r/LLM_PV-C04C/experiments/
>
> Experiments summary:
>
> Following the reviewers’ suggestions we added the following.
>
> (exp 1) LLM-PV with alternative proposers, including Gemini 3.1 Pro, Qwen3-30B-A3B-Thinking, DeepSeek-Coder-33B-Instruct, and OLMo-3-32B-Instruct + LLM-PV with GPT-5-Thinking without tool use
>
> (exp 2) GPT-5-Thinking in direct ICL form.
>
> (exp 3) MLAgentBench’s research agent using GPT-5-Thinking as the backbone model.
>
> (exp 4) An experiment to measure the training and testing runtime of different methods
>
> (exp 5) LLM-PV with 4 randomly generated prompts (were generated to be diverse using GPT-5)
>
> Conclusions: (1) the same backbone in direct ICL and MLAgentBench still underperforms LLM-PV, (2) different frontier agentic models also perform strongly within LLM-PV, (3) weaker or non-agentic variants degrade substantially, (4) LLM-PV is fairly robust to the prompt style, (5) the runtime of LLM-PV is not dramatically worse than the other methods. Together, these controls suggest that the gain is not explained by backbone strength alone, but by the combination of a sufficiently capable agentic proposer and the propose-execute-verify framework.
>
> > Reviewer: In experiments, the proposed method is compared with baseline LLM-based methods. Although the experimental results seem promising, the selection of LLMs seems unfair, as only LLM-PV uses the most powerful LLM, GPT-5-thinking.
>
> Response: We thank the reviewer for raising this concern. We agree that it is important to separate the effect of the LLM-PV framework from the effect of model strength. To address this directly, we added GPT-5-Thinking in direct ICL form, so that both methods use the same frontier model but only LLM-PV uses the propose-and-verify loop. Despite this much fairer comparison, GPT-5-Thinking ICL still performs substantially worse than LLM-PV across the benchmark (exp 2). This suggests that the improvement is not explained solely by backbone strength, but by the propose-execute-verify framework. We also expanded the proposer ablation beyond GPT-5. Gemini 3.1 Pro also performs strongly within the same LLM-PV pipeline, showing that the method is not tied to a single model family. By contrast, Qwen3-30B-A3B-Thinking, DeepSeek-Coder-33B-Instruct, and OLMo-3-32B-Instruct perform much worse overall, and GPT-5-Thinking without tool use also degrades markedly (exp 1). These results suggest that LLM-PV is not tied to GPT-5 specifically, but does rely on a sufficiently capable agentic proposer with tool-use capabilities. We will clarify this distinction in the revision.
>
> > Reviewer: Why do the authors use GPT-5-thinking only for the LLM-PV? How would the results change if we used a different model?
>
> Response: We used GPT-5-Thinking for LLM-PV to test whether a frontier agentic LLM can serve as an effective proposer in the propose-and-verify framework. To make the comparison fairer, we now also evaluate GPT-5-Thinking in direct ICL form, and it still performs substantially worse than LLM-PV. We also tested LLM-PV with different proposers. Gemini 3.1 Pro also performs strongly within the same framework, while Qwen3-30B-A3B-Thinking, DeepSeek-Coder-33B-Instruct, and OLMo-3-32B-Instruct perform much worse overall, and GPT-5-Thinking without tool use degrades markedly (exp 1). Thus, the effect is not specific to GPT-5 alone, but it does depend on having a sufficiently capable agentic proposer, rather than just any reasoning model.

---

> > ### Author Rebuttal · Reviewer_1nTN · 2026-04-04
> >
> > Thank you for providing the additional experimental results. Based on these results, it appears that LLM-PV is not effective for models other than GPT-5 Thinking (with tools) and Gemini 3.1 Pro. For this task, the ability to use tools seems more important than the training method.
> >
> > Does the additional ICL results (EXP 2) with GPT-5 thinking use tools? If tools are not used in ICL, then comparing GPT-5-ICL and LLM-PV (with tools) would be unfair since these models have different capabilities.
> >
> > Could you please clarify specifically which configurations utilize external tools?

---

> > > ### Author Response · Authors · 2026-04-05
> > >
> > > Thank you for the helpful follow-up. To address this more directly, we reran the ICL comparison using GPT-5-Thinking with tool-use and added a second ICL comparison using Gemini 3.1 Pro-Thinking with tool-use. In both cases, performance remains substantially below LLM-PV instantiated with GPT-5-Thinking and Gemini 3.1 Pro-Thinking. We therefore agree that tool-use is an important ingredient, but our results suggest that tool access alone does not explain the gain of LLM-PV over direct ICL. We also added another ablation with AIDE-ML (similar to MLAgentBench) with GPT-5-Thinking with tool-use, which also underperforms against LLM-PV.
> > >
> > > | Tasks | ICL: GPT 5 Thinking without Tools | ICL: Gemini 3.1 Pro Thinking with Tools | ICL: GPT 5 Thinking with Tools | MLAgentBench with GPT 5 Thinking with Tools | AIDE-ML with GPT 5 Thinking with Tools | LLM-PV: GPT 5 Thinking with Tools |
> > > | :--- | :--- | :--- | :--- | :--- | :--- | :--- |
> > > | FP | 82.0 | 48.0 | 99.0 | 53.6 | 49.5 | **100** |
> > > | FHP | 66.0 | 54.0 | 88.0 | 51.0 | 49.9 | **100** |
> > > | Pattern1 | 55.0 | 83.0 | 51.0 | 33.4 | 50.3 | **100** |
> > > | Pattern2 | 41.0 | 57.0 | 51.0 | 60.9 | 50.9 | **100** |
> > > | 3-P | 47.0 | 50.0 | 75.0 | 32.8 | 49.5 | **100** |
> > > | 10-P | 47.0 | 45.0 | 79.0 | 50.1 | 49.2 | **100** |
> > > | IsPal | 79.0 | 100.0 | 91.0 | 31.6 | 50.0 | **100** |
> > > | Dyck-2 | 57.0 | 51.0 | 59.0 | 52.1 | 50.5 | **100** |
> > > | CAP | 51.0 | 51.0 | 46.0 | 50.0 | 50.2 | **100** |
> > > | isPrime | 55.0 | 52.0 | 62.0 | 32.6 | 55.5 | **100** |
> > > | isPrime+47 | 47.0 | 56.0 | 56.0 | 33.1 | 49.7 | **80.6** |
> > > | CycleCount (seq_len=300) | 43.0 | 50.0 | 51.0 | 49.1 | 50.0 | **96.9** |
> > > | SHA | 44.0 | 45.0 | 46.0 | 49.9 | **50.8** | 50.1 |

---

### Official Review · Reviewer_AskF · 2026-03-13

**Soundness:** 2
**Presentation:** 3
**Significance:** 1
**Originality:** 2
**Overall Recommendation:** 4
**Confidence:** 3

**Summary:**

This paper studies the problem of sample- and computation-efficient program induction through empirical risk minimization (ERM). The authors highlight a fundamental tradeoff. On one hand, ERM can be highly sample efficient, requiring a number of samples that is logarithmic in the symbol alphabet and linear in the program length. However, naïvely implementing ERM via exhaustive program enumeration leads to a search that is exponential in the program’s description length, making it computationally infeasible. On the other hand, gradient-based learning methods are computationally tractable, often scaling linearly with the size of the training data, but they can require exponentially many samples to learn certain program families with short descriptions, such as parity or cryptographic functions.
The goal of the paper is to overcome this tradeoff by retaining the sample efficiency of ERM while avoiding the computational cost of exhaustive search. To achieve this, the authors propose an approach called LLM-PV (propose-and-verify). In this framework, a large language model provides a proposal distribution over candidate programs, which biases the search toward plausible solutions. Each proposed program is then evaluated on a held-out validation set, and the highest-scoring candidate is selected. Importantly, the validation feedback is used only for scoring and selection, not for updating the proposal distribution, and no gradient-based training is performed during the search.
Empirically, the authors show that the LLM-PV approach can successfully recover the correct underlying program and generalize beyond the training sequence length. In contrast, several alternative approaches, including classical machine learning models, transformers trained with SGD, in-context learning, and fine-tuning baselines, fail to achieve similar out-of-distribution generalization.

**Compliance With Llm Reviewing Policy:**

Affirmed.

**Final Justification:**

The additional baselines included during the rebuttal strengthen the submission and make it worthy of publication. That said, the paper would still benefit from a broader set of baselines and more extensive results. Nonetheless, the current evaluations appear sufficient to support the core claims.

**Key Questions For Authors:**

1. **Frontier model baselines.** Have you evaluated a strong frontier model (e.g., GPT-5-Thinking) directly in an in-context learning (ICL) setting as a baseline? Such a comparison would help determine how much of the performance gain comes from the propose-and-verify framework itself versus the capabilities of the underlying model.
2. **LLM-PV with weaker open-source models.** What happens when LLM-PV is instantiated with a smaller or open-source model, such as a Qwen variant, instead of GPT-5? Reporting these results would clarify how much the method depends on access to a very strong proposer.
3. **Accounting for pretraining in sample efficiency claims.** The paper argues that LLM-PV is sample efficient relative to SGD-based learners. However, the proposer relies on a heavily pretrained LLM whose training corpus may include vast amounts of data, potentially including synthetic data that overlaps with the task distribution. How should the sample efficiency claims be interpreted when the pretraining data is not accounted for?

**Limitations:**

yes

**Strengths And Weaknesses:**

### Strengths

1. **Strong theoretical grounding.** The paper provides rigorous analysis of the proposed method and situates it within the broader tradeoff between program enumeration and SGD-based learning, including PAC-style sample complexity considerations.
2. **Reasonable experimental coverage.** The evaluation includes a diverse set of tasks and multiple baselines, enabling comparison across several program induction approaches.
3. **Interpretability of the search process.** The program search procedure is transparent, and the use of chains of thought provides insight into how candidate programs are generated and refined.
4. **Improved generalization behavior.** The LLM-PV approach demonstrates stronger generalization beyond the training sequence length compared to several LLM-based baselines.

### Weaknesses

1. **Limited methodological novelty.** The propose-and-verify framework itself is conceptually simple and closely related to prior work using LLMs for program synthesis and programming-by-example (PBE) [1]. In particular, [1] shows that LLMs can act as strong proposers and can be further improved via synthetic data generation. As a result, the central message of the paper may appear incremental: using a strong frontier LLM as a proposal generator with rejection-style verification can outperform weaker models trained with fine-tuning, in-context learning, or classical ML methods.
2. **Potentially unfair experimental comparisons.** The main results compare GPT-5-Thinking (used in LLM-PV) against significantly smaller open-source models for the fine-tuning and ICL baselines. A more controlled comparison would evaluate GPT-5-Thinking under an ICL setting directly against LLM-PV, or alternatively run LLM-PV using a smaller open-source model and report those results.
3. **Dependence on a powerful pretrained model.** The effectiveness of LLM-PV relies on access to a strong pretrained LLM. This implicitly assumes the availability of a model trained on a massive pretraining corpus, which complicates claims about sample efficiency. Because the pretraining data for GPT-5 is not publicly known, it is difficult to assess whether the approach is truly more sample efficient than SGD-based learning. Using a fully open-source model such as OLMo [2], where the training data and process are transparent, would make the evaluation more scientifically grounded. Additionally, from a theoretical perspective, the distinction between LLM-PV and SGD-based learning is somewhat blurred, since the underlying LLM proposer is itself trained with SGD.
4. **Writing could be more concise in places.** While the paper is generally well written, some sections are dense and verbose. For instance, the abstract is unusually long and could likely be shortened without losing key information.

**References**
[1] Li, Wen-Ding, and Kevin Ellis. "Is Programming by Example Solved by LLMs?." The Thirty-eighth Annual Conference on Neural Information Processing Systems.
[2] OLMo, Team, et al. "2 OLMo 2 Furious." arXiv preprint arXiv:2501.00656 (2024).

---

> ### Author Rebuttal · Authors · 2026-03-31
>
> We thank the reviewer for their thoughtful and constructive feedback.
>
> New experiments: https://anonymous.4open.science/r/LLM_PV-C04C/experiments/
>
> For a summary see the first comment to reviewer 1nTN.
>
> > Reviewer: Limited methodological novelty with respect to PBE
>
> Response: We agree that our method overlaps conceptually with prior PBE and LLM-guided synthesis work. Our contribution is not a new search primitive, but a different learning setting: i.i.d. supervised learning with an explicit generalization objective, where candidate programs are selected by held-out validation error rather than exact consistency with the observed examples. This distinction matters in our regime, where standard baselines often fit the training data yet fail to generalize reliably, even at larger scales (Fig. 4), while exhaustive ERM over short programs is statistically attractive but computationally infeasible. LLM-PV aims to make that ERM-style route more practical by using a pretrained LLM only as a proposal mechanism, while retaining execution-based verification and validation-based selection.
>
> > Reviewer: [1] already shows that LLMs can be strong proposers and can be improved via synthetic data generation.
>
> Response: We agree there is conceptual overlap with Li and Ellis (2024), but it is narrower than the reviewer suggests. Their work studies classical PBE and develops a template-driven synthetic-data and fine-tuning pipeline: from a manually constructed seed set, it generates synthetic programs and inputs, executes them to create I/O pairs, trains an inference network $q_{\theta}(\rho \mid X,Y)$, and further adapts it with a wake-sleep style loop. Their prompting baseline also selects by exact consistency. By contrast, our setting is supervised ERM-style learning with held-out validation-based model selection over a discrete program class. We use the LLM only as a proposal prior, without domain-specific synthetic task generation or fine-tuning an inference network on $(\rho, X, Y)$ triples. Li and Ellis is thus relevant related work, but not a drop-in baseline for our benchmark.
>
> > Reviewer: Incremental central message: using a strong frontier LLM as a proposal generator with rejection-style verification can outperform weaker models
>
> Response: Empirically, this is not what our results show. The new controls sharpen the point: GPT-5-Thinking ICL (exp 2) and MLAgentBench (exp 3) still underperform LLM-PV, so the gain is not explained simply by using a stronger model. At the same time, a different frontier agentic model, Gemini 3.1 Pro, also performs strongly within LLM-PV, whereas weaker or non-agentic variants degrade substantially (exp 1). This suggests a more specific conclusion: what matters is not backbone strength alone, but a sufficiently capable agentic proposer within a PV loop. More broadly, standard learning methods still fail to recover these structured rules as effectively as LLM-PV, even as the training set grows.
>
> > Reviewer: Comparing LLM-PV and other methods using the same backbone model
>
> Response: We now compare against GPT-5-Thinking used directly for ICL, as well as MLAgentBench, an LLM-guided iterative pipeline-design baseline, and both perform substantially worse than LLM-PV (exp 3). We also instantiate LLM-PV with weaker open-source proposers to test sensitivity to proposer quality. Together, these controls show that the gain is not simply from “using GPT-5,” but from combining a sufficiently capable agentic proposer with the propose-execute-verify framework.
>
> > Reviewer: The method depends on a powerful pretrained model, which complicates sample-efficiency claims
>
> Response: Our sample-efficiency claims concern downstream labeled-sample efficiency given a fixed pretrained model, as in fine-tuning and ICL, not end-to-end efficiency including pretraining. The relevant question is therefore which adaptation mechanism works best in the low-data regime. The same-backbone GPT-5-Thinking ICL control is especially informative: it uses the same pretrained model as LLM-PV yet performs substantially worse. Open-model proposers also perform substantially worse than GPT-5-Thinking and Gemini 3.1 Pro (exp 1), indicating that proposer quality matters in practice.
>
> > Reviewer: Writing could be more concise in places
>
> Response: We will streamline the writing, shorten the abstract, and remove redundancy where possible.
>
> > Reviewer: How should sample-efficiency claims be interpreted given unknown pretraining data?
>
> Response: Our claims are about downstream labeled-sample efficiency given a fixed pretrained model, not total-data sample complexity including pretraining. The relevant comparison is therefore between adaptation mechanisms under the same or similar pretrained prior. The same-backbone GPT-5-Thinking ICL control makes this especially clear, since it uses the same pretrained model yet performs substantially worse than LLM-PV. We will make this explicit in the revision.

---

> > ### Author Rebuttal · Reviewer_AskF · 2026-04-04
> >
> > I thank the reviewers for providing additional experimental results, which I found informative. However, I would like to see the ICL vs. LLM-PV comparison for at least one additional model, preferably an open-source model, to better assess whether the proposed method is consistently a stronger adaptation scheme.
> >
> > Additionally, it is unclear whether ICL constitutes a fully fair baseline, as from what I can tell, it does not leverage execution-based feedback or validation in the same way as LLM-PV. While I understand that rebuttal time may not be sufficient to design and implement stronger baselines that incorporate execution feedback, such comparisons would be important for a more controlled and compute-matched evaluation of LLM-PV’s benefits.
> >
> > Given these issues, I choose to keep my current score for now.

---

> > > ### Author Response · Authors · 2026-04-05
> > >
> > > Thank you for the thoughtful follow-up. We agree that an additional same-backbone ICL vs. LLM-PV comparison is useful. To address this, we added two direct ICL comparisons, using GPT-5-Thinking with tool-use and Gemini 3.1 Pro-Thinking with tool-use, both of which are strong frontier backbones (see below). We find that both still underperform LLM-PV. This suggests that the improvement is not specific to a single model family.
> > >
> > > We also clarify that the current ICL setups do have access to execution through internal tool use, which allows the model to run tools and execute code behind the scenes. However, this is still different from LLM-PV, where the generated code is explicitly executed and validated within a controlled propose-and-verify loop. To address this fairness concern more directly, we additionally compare against more direct execution-based baselines, including MLAgentBench with GPT-5-Thinking with tool-use and AIDE-ML with GPT-5-Thinking with tool-use (see below). Both methods make explicit use of execution feedback, yet both remain substantially below LLM-PV on most tasks in our benchmark. These results suggest that the advantage is not explained solely by access to execution feedback, but by the propose-execute-verify adaptation scheme itself in this structured low-data regime.
> > >
> > > Finally, our claim is conditional rather than universal. We do not claim that LLM-PV is a drop-in improvement for arbitrary backbones. In our ablations, weaker open models perform substantially worse, suggesting that backbone strength alone is not sufficient. Instead, the evidence points to a different key ingredient: the ability to operate in a closed loop with tool use and explicit verification. A plain backbone is therefore not the right comparison object, much as theorem proving/coding typically depends on iterative multi-step reasoning and verification, capabilities that are present in frontier agents (e.g., GPT-5 and Gemini) but not in one-shot backbones.
> > >
> > > | Tasks | ICL: GPT 5 Thinking without Tools | ICL: Gemini 3.1 Pro Thinking with Tools | ICL: GPT 5 Thinking with Tools | MLAgentBench with GPT 5 Thinking with Tools | AIDE-ML with GPT 5 Thinking with Tools | LLM-PV: GPT 5 Thinking with Tools |
> > > | :--- | :--- | :--- | :--- | :--- | :--- | :--- |
> > > | FP | 82.0 | 48.0 | 99.0 | 53.6 | 49.5 | **100** |
> > > | FHP | 66.0 | 54.0 | 88.0 | 51.0 | 49.9 | **100** |
> > > | Pattern1 | 55.0 | 83.0 | 51.0 | 33.4 | 50.3 | **100** |
> > > | Pattern2 | 41.0 | 57.0 | 51.0 | 60.9 | 50.9 | **100** |
> > > | 3-P | 47.0 | 50.0 | 75.0 | 32.8 | 49.5 | **100** |
> > > | 10-P | 47.0 | 45.0 | 79.0 | 50.1 | 49.2 | **100** |
> > > | IsPal | 79.0 | 100.0 | 91.0 | 31.6 | 50.0 | **100** |
> > > | Dyck-2 | 57.0 | 51.0 | 59.0 | 52.1 | 50.5 | **100** |
> > > | CAP | 51.0 | 51.0 | 46.0 | 50.0 | 50.2 | **100** |
> > > | isPrime | 55.0 | 52.0 | 62.0 | 32.6 | 55.5 | **100** |
> > > | isPrime+47 | 47.0 | 56.0 | 56.0 | 33.1 | 49.7 | **80.6** |
> > > | CycleCount (seq_len=300) | 43.0 | 50.0 | 51.0 | 49.1 | 50.0 | **96.9** |
> > > | SHA | 44.0 | 45.0 | 46.0 | 49.9 | **50.8** | 50.1 |

---

### Official Review · Reviewer_R6Q1 · 2026-03-14

**Soundness:** 3
**Presentation:** 3
**Significance:** 3
**Originality:** 3
**Overall Recommendation:** 4
**Confidence:** 2

**Summary:**

The paper addresses the challenge of learning short programs from few examples while balancing sample efficiency and computational efficiency. The authors propose LLM‑PV, a propose‑and‑verify algorithm that uses a pretrained large language model as a data‑dependent proposal distribution over candidate programs and selects the program with the lowest validation error without performing gradient updates. Experiments on synthetic algorithmic tasks demonstrate that LLM‑PV recovers exact rules from about 200 labeled examples and generalizes to much longer input lengths, achieving perfect or near‑perfect accuracy on most tasks.

**Compliance With Llm Reviewing Policy:**

Affirmed.

**Key Questions For Authors:**

Q1. Can the propose‑and‑verify framework scale to more complex program synthesis tasks or real‑world domains beyond synthetic parity and primality tasks?

Q2. What is the computational cost (e.g., wall‑clock time) of sampling and verifying candidates with LLM‑PV, and how does it compare to training or in‑context learning?**

**Limitations:**

Yes

**Strengths And Weaknesses:**

+ The propose‑and‑verify paradigm combines pretrained LLM priors with ERM selection in a simple yet novel way, linking classical learning theory to modern generative models.
+ Theoretical analysis connects the propose‑and‑verify algorithm to classical ERM guarantees and derives a bound on the number of trials needed in terms of the proposal mass.
+ The paper is generally well written, motivating the compute–sample trade‑off and explaining the algorithm and analysis clearly.
+ LLM‑PV offers a promising alternative to gradient‑based training for algorithmic tasks, showing that LLM priors can guide efficient search to recover programs from few examples and generalize to longer inputs.

- The method relies on sampling from a proprietary GPT‑5 model; the sensitivity to model choice and prompt design is unexplored, leaving open whether smaller or open models would suffice.
- The computational cost of sampling and verifying candidates is not reported; large LLM inference costs might offset claimed efficiency.

---

> ### Author Rebuttal · Authors · 2026-03-31
>
> We thank the reviewer for their thoughtful and constructive feedback.
>
> New experiments: https://anonymous.4open.science/r/LLM_PV-C04C/experiments/
>
> For a summary see the first comment to reviewer 1nTN.
>
> > Reviewer: The sensitivity to model choice and would smaller models suffice
>
> Response: We now include proposer ablations beyond GPT-5, including Gemini 3.1 Pro, Qwen3-30B-A3B-Thinking, DeepSeek-Coder-33B-Instruct, OLMo-3-32B-Instruct, and GPT-5-Thinking without tool use (exp 1). Gemini also performs strongly within LLM-PV, while the weaker open models and the no-tool-use variant perform substantially worse overall. This suggests that the key requirement is not GPT-5 specifically, but a sufficiently capable agentic proposer with tool-use capabilities. In that sense, the relevant distinction is not simply proprietary vs. open, but whether the adaptation mechanism can actually unlock the needed search behavior in this regime.
>
> > Reviewer: Sensitivity to prompt choice
>
> To test prompt robustness, we reran LLM-PV with 4 random prompts on multiple tasks; performance remained unchanged (exp 5).
>
> P1 = Given a sequence of input vectors ({data_mode}, length {seq_len}) mapped to scalar binary outputs, extract the underlying relationship as a concise Python function f(x). The solution must be a direct logical or mathematical expression, not a machine learning model.
>
> P2 = Analyze the provided input vectors ({data_mode}, length {seq_len}) and their corresponding binary outputs to determine the governing logic. Express this logic as a short, deterministic Python function f(x) using mathematical or logical operations, avoiding trainable parameters.
>
> P3 =  Identify the mapping between the input vectors ({data_mode}, length {seq_len}) and binary scalar outputs. Represent this mapping through a concise, stateless Python function f(x) that relies purely on explicit mathematical or logical rules rather than learned weights.
>
> P4 = Discover the strict mathematical or logical rule that maps the input vectors ({data_mode}, length {seq_len}) to their binary outputs. Output a concise Python function f(x) that formally encodes this rule without relying on any trainable architecture.
>
> > Reviewer: What is the computational cost (e.g., wall‑clock time) of sampling and verifying candidates with LLM‑PV, and how does it compare to training or in‑context learning?
>
> Response: We now report explicit wall-clock comparisons (exp 4). For SGD and FT, we measure the runtime required to first fit the training data; for LLM-PV, we measure the runtime until the first iteration that fits the data. To make train-time and test-time costs more comparable, we normalize runtimes by the number of train or test samples, respectively. In LLM-PV, only a small number of executable candidates are evaluated per task, so the cost is dominated by a modest one-time search overhead. By contrast, ICL incurs repeated large-model inference at prediction time, while training-based baselines require substantially more optimization time. The resulting wall-clock comparisons support our main claim: LLM-PV is not always the cheapest method in absolute terms, but it offers a favorable compute-performance tradeoff in this structured low-data regime.
>
> > Reviewer: The computational cost of sampling and verifying candidates is not reported; large LLM inference costs might offset claimed efficiency.
>
> Response: Our claim is not that LLM-PV is inexpensive in absolute terms, but that it offers a more favorable compute-performance tradeoff than the alternatives we study in this structured low-data regime. In our main setup, LLM-PV evaluates only a small number of executable candidates, selected by validation error, and therefore pays a small one-time search cost per task. By contrast, ICL incurs repeated large-model inference at prediction time, while SGD and FT require substantial optimization compute yet still do not reliably recover the correct rule in this regime, even when the training set grows to 100k examples (Fig. 4). The wall-clock comparison below confirms this qualitative picture: LLM-PV incurs a modest upfront search cost, whereas ICL pays repeated inference cost. As per the paper, The other baselines require much more data.
>
> > Reviewer: Can the propose‑and‑verify framework scale to more complex program synthesis tasks or real‑world domains beyond synthetic parity and primality tasks?
>
> Response: We see this as a promising direction, but not as a claim established by the current paper. Our experiments focus on compact structured tasks, which are a natural testbed for propose-and-verify. More broadly, we believe the framework could extend to richer synthesis problems and some real-world structured domains through iterative propose-verify-refine updates over larger batches and more expressive candidate programs. We will clarify in the revision that the current paper establishes the method in structured low-data regimes, while broader scaling remains future work.

---

### Decision · Program_Chairs · 2026-04-30

**Decision:**

Accept (regular)

**Comment:**

The paper, focusing on short program synthesis, aims at the best out of both worlds: Empirical Risk Minimization (ERM) and gradient-based approach, where the former approach relies on sufficiently sampling the search space and the latter one is poorly suited to structured spaces.
The proposed approach combines an LLM, determining an appropriate distribution on the program space (biased toward short programs, according to the expert's priors about the desired solutions) , and a propose-and-verify (PV) procedure that retains the best out of $k$ samples of the distribution according to the error on a validation set.

The reviews and rebuttal convincingly establish some points:
* extensive comparisons and lesion studies establish that a good performance requires both i) a sufficiently powerful agentic LLM (GPT-5 or Gemini); ii) the PV procedure;
* the computational cost is on par with other LLM-based approaches

A last issue concerns the actual novelty of the approach, compared to the programming-by-example (PBE, Li and Ellis 2024).  The authors argue that their approach "shows that foundation-model-guided search can itself serve as a viable machine-learning mechanism for ERM over structured hypothesis classes".
I understand that the LLM is used to define an appropriate, general distribution on the target structured space; this distribution, made available, implies that simple and straightforward procedures (try and select) are now operational to solve the program synthesis task. This indeed shifts the usual learning lens.

The main question imo concerns the limitations of the approach: please indicate what happens for target programs with increasing length.